# Adaptive diagnostic reasoning framework for pathology with multimodal large language models

Yunqi Hong[1], Kuei-Chun Kao[1], Liam Edwards[2], Nein-Tzu Liu[3], Chung-Yen Huang[4], Alex Oliveira-Kowaleski [5], Cho-Jui Hsieh[1] ✉ & Neil Y. C. Lin [2,6,7,8] ✉

## Abstract

**Background** Artificial intelligence enhances pathology screening efficiency, yet clinical adoption remains limited because most systems operate as opaque black boxes. We aim to resolve this opacity by establishing a framework that generates transparent, evidence-linked reasoning to support diagnostic auditing.

**Methods** We present a framework that shifts off-the-shelf multimodal large language models from passive pattern recognition to active diagnostic reasoning. Using small labeled subsets from breast and prostate cancer datasets, we employ a two-phase self-learning process to derive diagnostic criteria without updating model weights. We integrate expert feedback from board-certified pathologists to ensure the generated descriptions align with established medical standards.

**Results** Here we show that our framework produces audit-ready rationales while achieving over 90% accuracy in distinguishing normal tissue from invasive carcinoma. Beyond binary classification, the model effectively differentiates complex subtypes like ductal carcinoma in situ by autonomously identifying hallmark histological features, including nuclear irregularities and structural disruption. These computer-generated descriptions closely match expert assessments. Our approach delivers substantial performance gains over conventional baselines and adapts effectively across diverse tissue types and independent foundation models.

**Conclusions** By uniting visual understanding with reasoning, our framework provides a promising approach for clinically trustworthy artificial intelligence. This framework helps bridge the gap between opaque classifiers and auditable systems, suggesting a viable path toward evidence-linked interpretation in medical workflows.

## Plain language summary

Computer programs can assist pathologists in examining tissue samples to detect cancer, but these tools often fail to explain how they reach their conclusions. We developed a new method to make this process transparent. A key innovation of our approach is that the system automatically learns the specific criteria for diagnosing cancer on its own by analyzing tissue images, rather than relying on pre-programmed rules. We tested this on breast and prostate samples, and the system successfully identified cancer while providing clear written descriptions of cell abnormalities that matched the assessments of specialist doctors. This research demonstrates that computers can be accurate and provide understandable reasoning, allowing doctors to verify automated results and ensuring safer evaluations for patients.

Over the past two decades, diagnostic pathology has been transformed by increasingly sophisticated artificial intelligence (AI) and computer vision methods[1–3]. Early efforts focused on extracting handcrafted features such as texture, color, and shape, but today deep neural networks dominate the field, powering a wide range of clinical and research applications[4–6]. These advances are reshaping pathology laboratories on multiple fronts.

Automated systems can now scan digitized histopathological slides at scale, rapidly identifying tumors and enabling pathologists to focus on urgent cases[7,8]. Algorithms that quantify immunohistochemical (IHC) markers further improve prognostic accuracy and reproducibility[9,10]. At the same time, AI-powered quality control dashboards monitor staining variability, providing near real-time feedback to maintain laboratory standards[11–13].

[1]Computer Science Department, University of California, Los Angeles, CA, USA. [2]Mechanical and Aerospace Engineering Department, University of California, Los Angeles, CA, USA. [3]Department of Pathology, Tri-Service General Hospital, National Defense Medical Center, Taipei, Taiwan. [4]Department of Pathology, National Taiwan University Hospital, Taipei, Taiwan. [5]Department of Pathology, David Geffen School of Medicine, University of California, Los Angeles, CA, USA. [6]Bioengineering Department, University of California, Los Angeles, CA, USA. [7]Institute for Quantitative and Computational Biosciences, University of California, Los Angeles, CA, USA. [8]Jonsson Comprehensive Cancer Center, University of California, Los Angeles, CA, USA. ✉e-mail: chohsieh@cs.ucla.edu; neillin@g.ucla.edu

Together, these innovations are moving pathology from subjective interpretation to algorithmic precision, redefining the pathologist's role in the future of medicine.

Despite substantial progress in classification accuracy and diagnostic efficiency, most current models remain black boxes that cannot explain predictions in a way that reflects the reasoning of human experts[14,15]. In clinical practice, trust depends on explanations that align with evidence-based reasoning, both for pathologists and for regulatory oversight[16,17]. Traditional interpretability methods, such as attention maps[18,19], concept-level explanations[20,21], and data attributions[22] provide only partial post-hoc insights. More recent approaches in biomedical imaging[23,24] often demand densely annotated datasets or generate coarse saliency maps that lack the granularity needed for auditability and clinical trust[25]. Without transparent and audit-ready justifications, even highly accurate AI models cannot be reliably integrated into pathology, where every diagnostic call must be defensible to tumor boards and regulatory bodies.

In parallel, recent breakthroughs in multimodal large language models (MLLMs)[26-32] demonstrate strong image understanding and emerging reasoning skills. LLM prompting and in-context learning[23,33-39] can enable medical image diagnosis[40-42]. But many current systems often provide explanations that are unstructured, weakly grounded in morphological features, or inconsistent across runs[43]. Such limitations reduce their suitability for clinical audit and trust. Even so, MLLMs remain a compelling foundation for addressing the central challenge of trustworthy AI in pathology, as recent models support higher-resolution visual inputs and longer multimodal context that can be harnessed to structure stepwise, feature-grounded reasoning[26,28].

Building on these recent advances, we propose RECAP-PATH (REasoning and Classification via Automated Prompting in PATHology images), a framework designed to facilitate clinical verification that leverages MLLMs to deliver accurate diagnoses with pathologist-aligned justifications. The system articulates the morphologic evidence supporting each diagnostic decision and jointly evaluates this evidence with the image to produce the final prediction. At its core is a "think-and-speak" stage, where an MLLM, guided by a description-generation prompt enriched with histopathological features, generates a detailed account of the image and the diagnostic significance of each feature. To identify diagnosis guidelines that yield accurate and task-specific rationales, we iteratively refine the description-generation prompt using diagnostic feedback derived from discrepancies between predictions and ground truth labels, leveraging the error reflection ability discovered in recent prompt optimization work[44-47]. This refinement progressively enhances clarity and diagnostic relevance without finetuning model weights. The framework is highly efficient, requiring as few as 100 labeled examples, no white-box access, and no additional MLLM retraining, while operating entirely through standard API calls. To the best of our knowledge, this is the first approach to jointly optimize visual description generation with prompt-based optimization to deliver audit-ready, human-interpretable diagnostic outputs.

## Methods
### Datasets
All data used in this study were obtained from publicly available, anonymized BRACS, BACH, and SICAPv2 histopathology image datasets and were used solely for research purposes. All usage complied with the terms of use of the respective data platforms and licenses. Institutional Review Board (IRB) approval was not required for this study, as all datasets used are publicly available and fully anonymized. Specifically, our experiments are conducted on the following histopathology image datasets:

- **BRACS**[48] (BReAst Carcinoma Subtyping) is a dataset of hematoxylin and eosin (H&E) stained histopathological images designed for automated detection and classification of breast tumors. It consists of 4539 labeled Regions of Interest (RoIs) extracted from 387 whole-slide

images (WSIs) collected from 151 patients. The RoIs vary in size and can exceed 4000 by 4000 pixels. In our experiments, we consider two classification tasks: (1) distinguishing cancerous tissue from non-cancerous tissue (i.e., Normal (N) vs. Invasive Carcinoma (IC)), and (2) classifying two cancer subtypes: Ductal Carcinoma In Situ (DCIS) and Invasive Carcinoma (IC). Each group includes hundreds of training samples and approximately 80 test samples. For optimization, we use only 30 training samples per category, which is sufficient to achieve strong performance with reduced computational cost.

- **BACH**[49] (Breast Cancer Histology) dataset is a public collection of high-resolution microscopy images designed for the classification of breast cancer subtypes. The complete dataset contains 400 RGB images in TIFF format, equally distributed across four clinically relevant classes: normal, benign, in situ carcinoma, and invasive carcinoma, with 100 images per class. Each image has a resolution of $2048 \times 1536$ pixels with a pixel scale of 0.42 μm, and was annotated by expert pathologists. For our experiment, we focused on a binary classification task to distinguish between the Normal and Invasive carcinoma in these two categories. We combined the 100 images from each of these two classes and performed a random 60/40 split while maintaining label balance. This resulted in a training set of 120 images (60 Normal, 60 Invasive) and a test set of 80 images (40 Normal, 40 Invasive) for the experiment.

- **SICAPv2**[50] dataset is a comprehensive collection of 18,783 10× magnified histological prostate images from 155 patients, each with a resolution of $512 \times 512$ pixels and detailed per-pixel annotations under Gleason grading (GG), which is the main diagnostic and evaluative tool for prostate cancer in clinical practice. The dataset is originally graded into four classes: Non-cancer (NC), G3, G4, and G5. For our experiment, we focused on a binary classification task to distinguish between Non-cancer and Cancer. To accomplish this, we first consolidated the G3 (characterized by atrophic, highly differentiated, dense gland areas), G4 (characterized by cribriform, pathological, large confluent, papillary glands), and G5 (characterized by individual cells, no light cavity forming, pseudorosette cell nests) grades into a single "Cancer" category. From this structure, we created a balanced training set by randomly sampling 100 images from the Non-cancer class and 100 images from the combined Cancer class. For robust evaluation, we then constructed a test set by randomly sampling 644 images from the remaining non-cancer pool and 642 images from the remaining Cancer pool, resulting in a nearly balanced test set of 1286 images.

### Automatic prompt optimization with description generation
**Image description generation and diagnosis.** Let $(x, y)$ denote an image-label pair from the dataset $\mathscr{D} = \{(x_i, y_i)\}_{i=1}^N$, and let $p$ be a classification prompt associated with the dataset. In a standard zero-shot setting, a general-purpose MLLM predicts the label directly from the image using the prompt, formulated as $\hat{y} = \text{MLLM}(x, p)$.

Our method introduces a two-step prediction pipeline to enhance interpretability and diagnostic accuracy, as illustrated in Fig. 1A. Specifically, we incorporate a description generation prompt $q$ to first guide the model to generate a structured textual description of the input image. This process is denoted as $s = \text{MLLM}(x, q)$. The generated description $s$ can provide additional contextual cues to the model. In the second step, we concatenate the classification prompt $p$ with the generated description $s$ and pass them, along with the image $x$, to the MLLM for final prediction. This step is represented as $\hat{y} = \text{MLLM}(x, [p, s])$.

**Automated prompt refinement for image descriptions.** Shown in Fig. 1B, we utilize a small number of labeled training data to optimize the generation prompt $q$, so that more precise and informative descriptions could be generated to make the diagnosis more reliable. We formalize an optimization objective based on the prompt diversity or training set prediction accuracy to evaluate the effectiveness of each candidate generation prompt. The algorithm is illustrated in Algorithm 1.

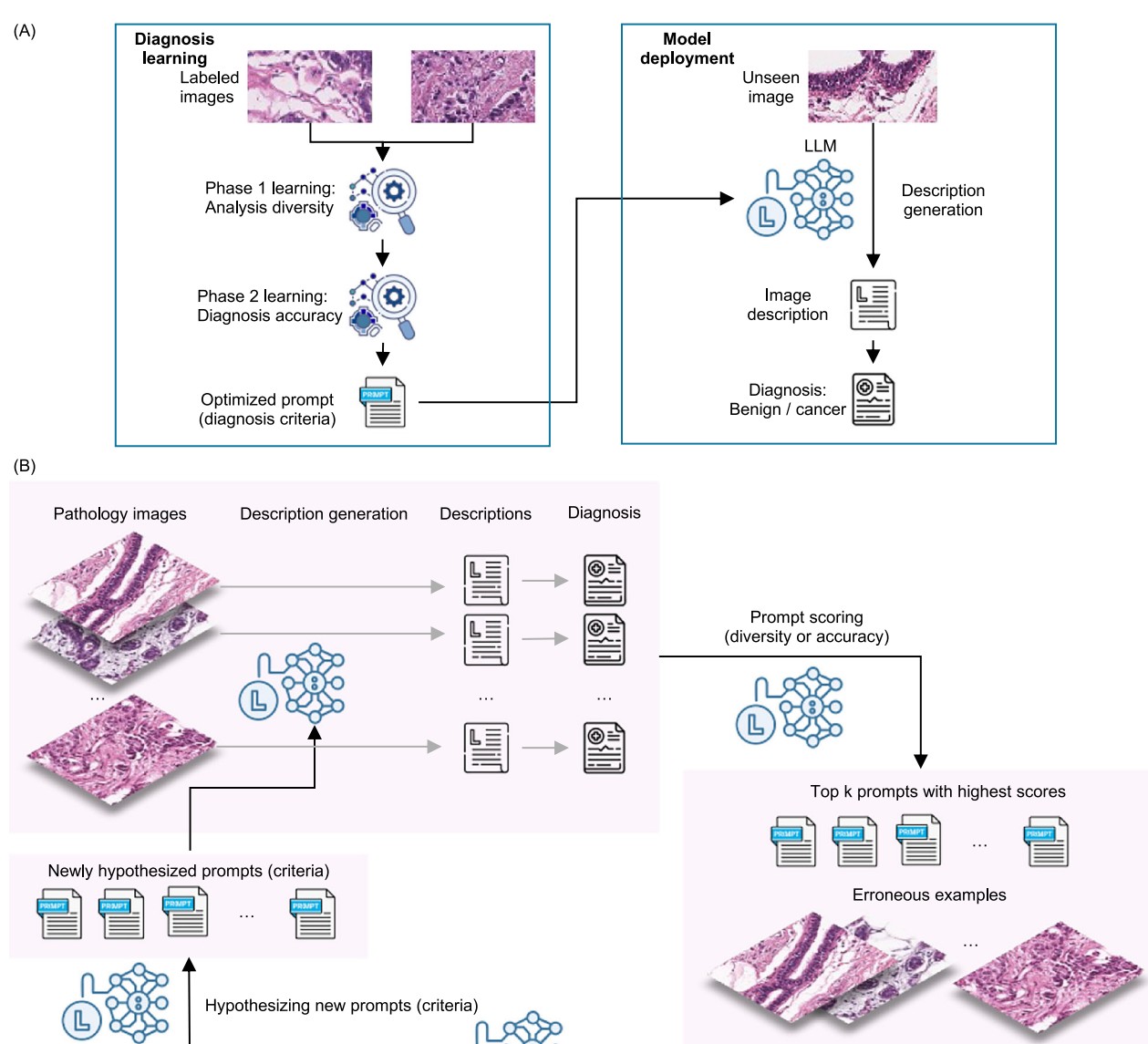

**Fig. 1 | Framework of RECAP-PATH. A** Overview of the RECAP-PATH learning framework and deployment pipeline. Using a small set of labeled pathology images (left), RECAP-PATH conducts a two-phase diagnostic learning process that yields an optimized prompt encapsulating the diagnosis criteria. During inference on unseen pathology images (right), the model generates detailed image descriptions guided by the optimized criteria and then produces classification predictions informed by both visual features and textual descriptions. **B** Schematic of the automatic prompt refinement workflow. In each iteration, RECAP-PATH identifies error cases, prompts the model to reflect on failure modes, and generates revised prompts aimed at enhancing diagnostic accuracy and improving human-readable diagnostic rationales. Through this iterative, error-driven refinement, the framework produces prompts that are both clinically meaningful and performance-optimized. Representative examples of such prompts are shown in Fig. S1.

---

**Algorithm 1**. Automatic Prompt Optimization of Image Descriptions

**Require:** $q_0$: initial generation prompt, $p$: prediction prompt, $N$: iterations, $\mathscr{D}$: training dataset, $b$: top prompts retained per iteration, $l$: number of error examples for reflections, $S(\cdot)$: scoring function

1: $Q_0 \leftarrow \{q_0\}$ ▷ Initialize candidate prompt set

2: $\Gamma \leftarrow \{\gamma_0\}$, where $\gamma_0 = \{(x_i, s_i)|s_i = \mathrm{MLLM}(x_i, q_0), (x_i, y_i) \in \mathscr{D}\}$ ▷ Generate descriptions

3: **for** $t = 1$ to $N$ **do**

4: $Q_c \leftarrow Q_{t-1}$

5: **for** $q \in Q_{t-1}$ **do**

6: $J_{\mathrm{error}} = \{(x_i, y_i)|\hat{y}_i \neq y_i, \hat{y}_i = \mathrm{MLLM}(x_i, [p, s_i]), (x_i, s_i) \in \gamma, \gamma \in \Gamma, (x_i, y_i) \in \mathscr{D}\}$ ▷ Collect errors

7: $J_{\mathrm{error}}^i \subset J_{\mathrm{error}}$ for $i = 1, \ldots, l$ ▷ Sample a subset for each $i = 1, \ldots, l$

8: $G = \{g_1, g_2, \ldots g_l\} = \bigcup_{i=1,\ldots l} \mathrm{Reflect}(p, J_{\mathrm{error}}^i)$ ▷ Reflect on the errors

9: $H = \{h_1, h_2, \ldots h_l\} = \bigcup_{i=1,\ldots i} \mathrm{Modify}(p, g_i, J_{\mathrm{error}}^i)$ ▷ Modify prompts

10: $Q_c \leftarrow Q_c \cup H$

11: $\Gamma \leftarrow \Gamma \cup \{\{(x_i, s_i)|s_i = \mathrm{MLLM}(x_i, h), (x_i, y_i) \in \mathscr{D}\}|h \in H\}$ ▷ Generate descriptions for new prompts

12: **end for**

13: $\mathscr{S}_c = \{S(q)|q \in Q_c\}$ ▷ Evaluate prompts

14: $Q_t \leftarrow \{q \in Q_c|S(q) \geq \tau\}$, where $\tau$ is the $b^{\mathrm{th}}$ highest score in $\mathscr{S}_c$

15: **end for**
16: **Return** $q^* \leftarrow \arg\max_{q \in Q_N} S(q)$

**2-Phase optimization**. Our method consists of two optimization phases with different scoring functions $S(\cdot)$. In the first phase, we optimize for prompt diversity. We quantify this diversity using two components: the **terminology count** ($T(q_i)$), which is the number of biomedical terms in a prompt $q_i$, and **term uniqueness** ($U(q_i)$), the number of biomedical terms in $q_i$ that do not appear in any other prompt $q_j$ (where $j \neq i$). To ensure a balanced contribution, we normalize these counts by the maximum value observed across all prompts:

$$\widetilde{T}(q_i) = \frac{T(q_i)}{\max_j T(q_j)}, \quad \widetilde{U}(q_i) = \frac{U(q_i)}{\max_j U(q_j)}.$$

The final diversity score $D(q_i)$ is defined as the sum of these two normalized components:

$$D(q_i) = \widetilde{T}(q_i) + \widetilde{U}(q_i).$$

We set the diversity scoring function $S(q) = D(q)$ for this phase. This formulation ensures that both the richness of terminology and the uniqueness of each prompt contribute equally to the overall diversity measure.

In the second phase, we define the scoring function as the diagnostic accuracy on a training dataset $\mathscr{D} = \{(x_i, y_i)\}_{i=1}^N$, which is essentially the proportion of correctly classified samples:

$$S(q) = \text{Accuracy} = \frac{1}{N} \sum_{i=1}^N \mathbb{1}\left[\widehat{y}_i = y_i\right] = 1 - \frac{|J_{\text{error}}|}{|\mathscr{D}|},$$

where $\widehat{y}_i = \text{MLLM}(x_i, [p, s_i])$ is the model's prediction given image $x_i$ and description $s_i$ generated with prompt $q$, and $\mathbb{1}[\cdot]$ denotes the indicator function.

**Model and hyperparameter setup**
All the experiments in our study were conducted using Gemini 2.0 Flash[51], a state-of-the-art general-purpose MLLM released on February 5, 2025 (model string: "gemini-2.0-flash-001"), and OpenAI GPT-4o (model string: "gpt-4o-2024-05-13"). We set the temperature to 0.0 for deterministic classification predictions and 0.7 for more creative and informative image description generation. We set the number of top prompts retained per iteration $b$ to be 4, the number of error examples for reflections $l$ to be 4.

**Comparison methods and validation benchmarks**
To rigorously evaluate the performance, efficiency, and generalizability of our proposed framework, we conducted experiments using three distinct categories of baselines: (1) a standard classification model to establish a performance benchmark for diagnostic accuracy; (2) a state-of-the-art prompt optimization method to compare optimization efficiency; and (3) an open-source MLLM to validate the applicability of our framework beyond proprietary commercial models.

- **CLIP**[52] (Contrastive Language-Image Pretraining) serves as a fundamental vision-language baseline. It is trained by jointly learning image and text encoders to map both modalities into a shared embedding space using a contrastive objective. The model optimizes a symmetric cross-entropy[53] (InfoNCE) loss over batches of image-caption pairs to maximize the cosine similarity of matched pairs while minimizing it for non-matching ones. We utilized CLIP for its zero-shot classification capabilities, where the image embedding is compared directly to the embeddings of class-specific text descriptions without additional training.
- **APE**[54] (Automatic Prompt Engineering) represents a state-of-the-art in automated prompt optimization. APE frames prompt engineering as a black-box optimization problem where instructions are treated as

programs to be generated and selected by an LLM. The framework starts with a pool of candidate prompts generated by an LLM, evaluates them based on zero-shot task accuracy or similar metrics, and iteratively refines the selection. We used this baseline to compare our proposed method against a pure LLM-driven prompt search approach that does not explicitly leverage the structured visual description loop used in our pipeline.
- **Qwen2.5-VL**[55] serves as a representative baseline for MLLMs. It integrates a native dynamic-resolution Vision Transformer[56] with the Qwen2.5 language model. The architecture incorporates enhancements such as window attention, SwiGLU activations, RMSNorm[57], and Multimodal Rotary Position Embeddings[58] to efficiently model spatial and temporal information. We utilized the instruction-tuned variant, which is optimized for visual reasoning and instruction following, to evaluate performance when a single, unified model processes both visual and textual inputs without the explicit two-phase optimization we propose.

**Data analysis and visualization**
To generate the UMAP visualizations, we first embedded the generated image descriptions using Gemini text-embedding-004 (released on May 14, 2024), which projects each description into a 768-dimensional semantic space. We then applied UMAP[59] with Euclidean distance as the similarity metric to reduce the high-dimensional embeddings to a two-dimensional space for visualization.

**Expert blind evaluation survey**
To assess the clinical relevance and interpretability of RECAP-PATH, we conducted a blind evaluation survey in which three board-certified pathologists reviewed images and their corresponding descriptions generated by the MLLM. Each pathologist was presented with histopathology images alongside corresponding descriptions generated by the MLLM. They evaluated the accuracy, completeness, and clarity of the descriptions and provided structured feedback on diagnostic relevance. This expert feedback was incorporated into the optimization process, guiding refinements in prompt design and improving the quality of generated outputs. In addition, we performed a blinded comparative evaluation in which pathologists were asked to review descriptions generated with and without pathology-specific knowledge augmentation. Without being informed of the source, they rated each description on criteria such as diagnostic accuracy, interpretability, and usefulness for clinical decision-making.

## Results
### Two-phase prompt optimization enables accurate and rationale-based triage of invasive carcinoma
To implement RECAP-PATH's adaptive reasoning capabilities, we introduce an iterative two-phase description-based prompt optimization algorithm (Fig. 1A) that enables an MLLM to adapt and iteratively refine diagnostic rationales through feedback derived from discrepancies between predictions and ground-truth labels expressed in natural language. As illustrated in Fig. 1B, this prompt optimization workflow is designed to enhance both diagnostic accuracy and interpretability by guiding the MLLM to generate structured, morphologically grounded, pathologist-style descriptions. The process begins with baseline prompts that provide minimal diagnostic guidance, instructing the MLLM to describe visual observations of tissue images and produce corresponding diagnostic predictions. In each iteration, the model reflects on misclassifications, identifies error patterns and likely sources of diagnostic confusion, and proposes strategies for improvement. It then generates a diverse set of candidate prompts embedding domain-specific morphological criteria and explicit reasoning steps aligned with expert pathology practice. Candidate prompts are evaluated against labeled data based on diagnostic accuracy, with the top three retained for further refinement. Focusing on the optimization phase for clarity, we provide a representative example of the prompts used to guide

each step of the refinement process, spanning description generation, classification, error collection, feedback synthesis, and guideline formulation, in Fig. S1.

Our two-phase description-optimized prompting strategy is designed to improve diagnostic accuracy while fostering interpretable, expert-aligned reasoning. Unlike standard automatic prompt optimization methods that focus solely on performance metrics[44,45], our approach leverages the model's own diagnostic errors to guide prompt evolution and encourage diverse explanatory strategies. To achieve this, the process is divided into two complementary phases. **Phase 1 (Diversification)** prompts the MLLM to expand the semantic and conceptual scope of its diagnostic explanations. The model is encouraged to introduce novel terminology, explore alternative morphological perspectives, and apply varied interpretive lenses. This phase aims to maximize interpretive diversity, enabling the exploration of multiple diagnostic reasoning pathways without being constrained by immediate accuracy. **Phase 2 (Optimization)** then refines this pool of prompts by selecting those that enhance diagnostic performance while maintaining clinical coherence. Prompts that do not contribute meaningfully to accuracy are discarded, while those yielding the most effective and interpretable outputs are iteratively improved. Together, these phases create a structured pipeline that first explores diverse reasoning strategies and then converges on clinically high-performing explanations.

To demonstrate the capabilities of our framework, we first evaluate it on a clinically critical task: binary classification of normal versus invasive carcinoma in breast pathology images, a foundational distinction for diagnosis and prognosis. Consistent with our design principles, the two-phase optimization process exhibits distinct learning dynamics in each phase (Fig. 2A). In Phase 1, classification accuracy declines slightly as the model explores a wider range of diagnostic reasoning strategies that increase diversity but are not yet optimized for performance. Once Phase 2 begins, accuracy rises rapidly, with substantial gains appearing within the first few iterations. Convergence is typically reached in about six rounds, and the final prediction accuracy exceeds 0.9. This nonlinear trajectory indicates the effectiveness of decoupling interpretive exploration from performance optimization, resulting in a set of prompts that are both diagnostically accurate and clinically audit-ready.

To benchmark this performance, we established a supervised upper bar using a linear classifier trained on contrastive language-image pre-training (CLIP) embeddings[52], which achieved 92.5% accuracy. RECAP-PATH attains comparable results without requiring weight updates or white-box access, while providing pathologist-readable rationales. Additionally, an in-context learning baseline using manually curated clinical definitions yielded lower performance (86.0% for N/IC) than RECAP-PATH (Table 1, Fig. S2). We further benchmarked against Automatic Prompt Engineering (APE)[54], which achieved 88.0% (95% CI: 83.1–92.9%) and underperformed RECAP-PATH (91.3%). These comparisons indicate that neither static expert knowledge nor general-purpose instruction tuning is sufficient without the systematic, reasoning-driven alignment provided by our optimization.

To evaluate the effectiveness of prompt diversification, we quantified terminology diversity across successive rounds of optimization. Diversity was measured using two criteria: (i) the number of biologically relevant terms and (ii) the number of words unique to the current prompt relative to all previously generated prompts. As shown in Fig. 2B, diversity increased steadily during Phase 1, reflecting a successful expansion of the diagnostic reasoning space. In Phase 2, a modest reduction in diversity was observed as the optimization process shifted focus toward maximizing diagnostic accuracy. However, the final set of prompts maintained a substantially higher level of lexical and conceptual diversity, more than twice that of prompts generated without the diversification phase (Fig. 2C). While single-phase optimization (focused solely on accuracy) achieved comparable classification performance, the resulting diagnostic guidelines and descriptions were noticeably less structured and lacked the depth provided by diversified prompts (Fig. S3). These results show that our framework preserves diverse

diagnostic language while integrating it into high-performing prompts, supporting robustness against overfitting or narrow reasoning.

## Prompt evolution drives bias rebalancing and emergent diagnostic structure

To assess the impact of prompt refinement on diagnostic behavior, we tracked changes in the model's confusion matrix across both optimization phases (Fig. 2D). In the initial zero-shot setting, the MLLM exhibited a pronounced bias toward invasive carcinoma classifications, likely reflecting distributional priors from its pretraining data. Introducing a generic seed prompt slightly mitigated this bias, but it was only after completing the full two-phase prompt optimization that the confusion matrix reflected well-calibrated and accurate predictions. This progression shows that our framework mitigates model bias by guiding diagnostics toward evidence-based decisions. Moreover, the two-phase optimized diagnostic guidelines enhanced consistency, reducing classification variance by nearly half compared to the initial seed prompt (Fig. S4). This improved consistency was also evident in the corresponding image descriptions, which demonstrated substantially greater coherence and depth (Fig. S4).

We next evaluated whether our framework enhances diagnostic reasoning by analyzing the content and structure of the optimized prompt guiding the MLLM's predictions (Fig. 2E). The initial baseline prompt, which only requested discriminative characteristics, gradually evolved into well-structured and pathologically coherent guidelines through iterative optimization. It consistently invoked five hallmark dimensions of histopathologic assessment: invasion phenotype, tissue architecture, nuclear morphology, stromal context, and mitotic activity. Notably, these clinically salient criteria emerged without explicit supervision or handcrafted rule encoding. Together, these findings demonstrate that our two-phase description-optimized prompting framework induces reasoning patterns that mirror established pathology workflows, reinforcing its potential as an interpretable diagnostic tool.

To investigate how our prompt refinement method shapes the MLLM's alignment with diagnostic tasks, we analyzed the text embeddings of MLLM-generated image descriptions and visualized their distribution using Uniform Manifold Approximation and Projection (UMAP) (Fig. 2F). Descriptions generated from the initial baseline prompts produced highly overlapping clusters for normal tissue and invasive carcinoma, indicating limited discriminative capacity. As the prompts were iteratively refined, the embedding space progressively disentangled, ultimately forming two well-separated clusters corresponding to benign and malignant phenotypes. Interestingly, early descriptions of invasive carcinoma briefly diverged into two distinct semantic trajectories (intermediate panel in Fig. 2F) before merging into a single coherent group. This transient bifurcation suggests that the optimization process not only enhances semantic clarity but also captures intermediate variability on the path toward a unified diagnostic narrative. Taken together, these findings indicate that interpretability in our framework emerges naturally from the prompt optimization process rather than being imposed as a post hoc addition.

## MLLM descriptions enable precise histological interpretation

Although the two-phase description-optimized prompting process enhances interpretability and diagnostic performance, achieving clinically meaningful reasoning requires alignment with expert judgment. To address this, we integrated human-AI interaction into the optimization loop by soliciting feedback from three board-certified pathologists (Fig. 3A). These experts performed blinded evaluations of LLM-generated image descriptions, covering both normal and invasive carcinoma cases at different stages of the optimization process (example questionnaire shown in Fig. S5). Each description was rated for precision and histopathological accuracy, and these assessments directly informed subsequent prompt revisions (example feedback shown in Fig. S6). The MLLM was then instructed to incorporate expert-derived principles to improve diagnostic reliability. This process ensured that the model's reasoning was not only internally consistent but also aligned with established clinical standards. As shown in Fig. 3B, expert-

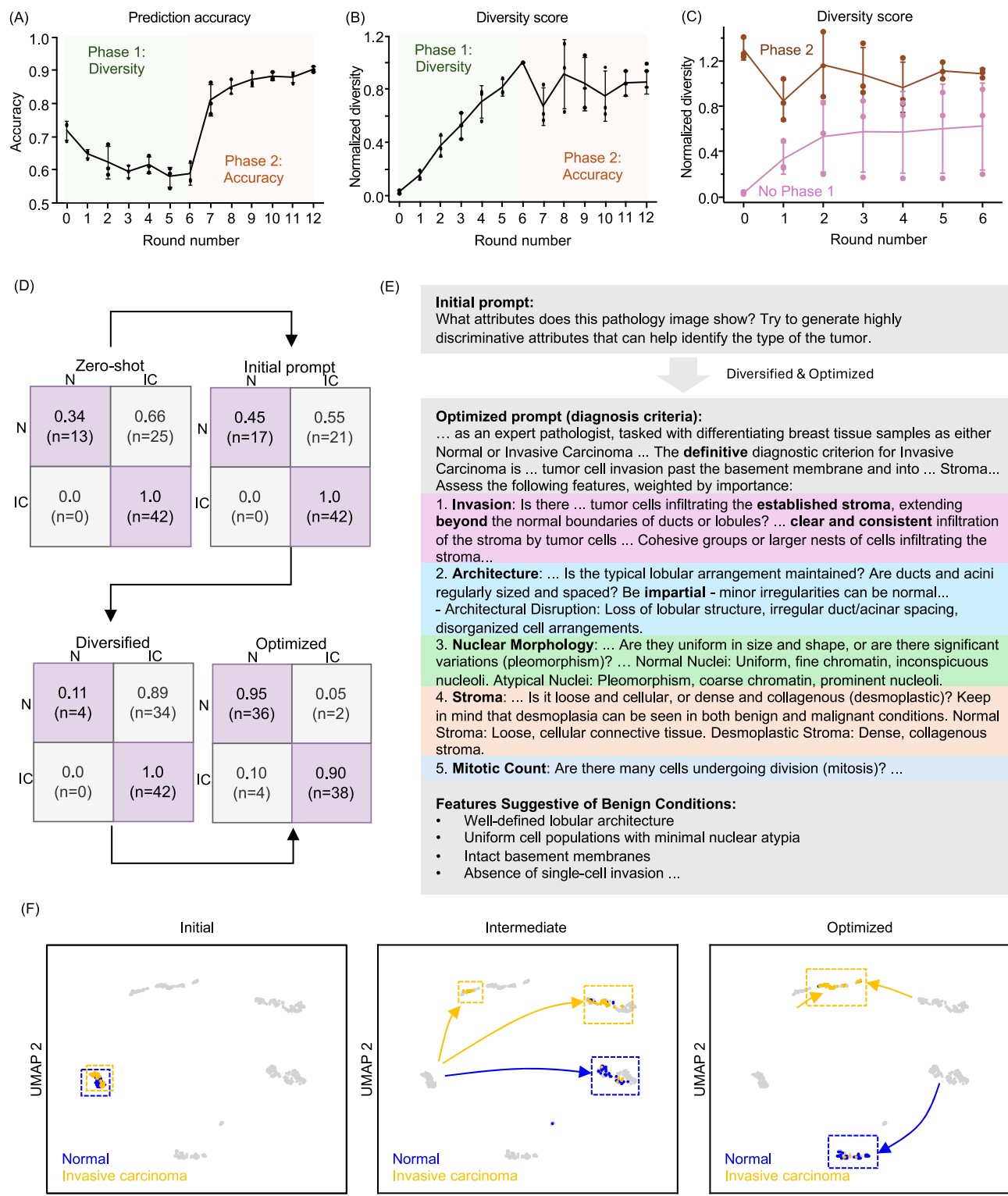

**Fig. 2 | Learning dynamics and prompt evolution in RECAP-PATH. A** Prediction accuracy over learning iterations. Accuracy decreases slightly during Phase 1 (diversification) as the model explores a broader range of diagnostic reasoning strategies. In Phase 2 (accuracy), accuracy increases rapidly, with substantial improvements achieved within a few iterations and convergence around six rounds. **B** Prompt diversity over learning iterations. Diversity steadily increases during Phase 1 as the goal is to have more diverse reasoning strategies. During Phase 2, diversity decreases slightly as prompts are refined for accuracy, but remains substantially higher than the starting point. **C** Impact of the diversification phase. Incorporating Phase 1 leads to significantly greater lexical and conceptual diversity in the final prompt compared to training

without diversification. **D** Evolution of the test confusion matrix. The model progresses from an initial zero-shot bias toward one category to a well-balanced, optimized diagnostic performance. **E** Example of prompt evolution. The initial seed prompt is generic and simple, while the final optimized prompt reflects a structured, clinically meaningful diagnostic framework. **F** UMAP visualization of the description embeddings. Gray points represent aggregate embeddings from all stages as a static reference. Initially, descriptions for the two classes overlap with poor separability. After optimization, descriptions form two well-separated clusters, demonstrating semantic disentanglement aligned with diagnostic categories. Error bars in **A–C** indicate standard deviation derived from $n = 3$ independent optimization experiments.

## Table 1 | Cross-model evaluation of RECAP-PATH on the BRACS dataset

| Task | Strategy | Gemini 2.0 Flash | GPT-4o |
|------|----------|------------------|--------|
| N/IC | Initial | 71.87 (69.26, 74.48) | 52.25 (46.20, 58.30) |
| | Optimized | 91.25 (90.29, 92.21) | 80.25 (77.95, 82.55) |
| | In-context Learning | 86.00 (84.70, 87.30) | – |
| DCIS/IC | Initial | 82.50 (79.69, 85.31) | 59.50 (57.73, 61.27) |
| | Optimized | 87.81 (84.83, 90.80) | 89.50 (86.68, 92.32) |
| | In-context Learning | 79.50 (78.11, 80.89) | – |
| N/DCIS/IC | Initial | 60.28 (55.63, 64.93) | 52.00 (47.70, 56.30) |
| | Optimized | 77.78 (76.40, 79.16) | 63.78 (60.18, 67.38) |
| | In-context Learning | 60.47 (59.28, 61.66) | – |
| N/IC | Optimized by Peer | 86.75 (83.72, 89.78) | 70.96 (68.32, 73.59) |
| (Cross-model) | (Train ≠ Infer) | Train: GPT-4o | Train: Gemini 2.0 |

Values represent classification accuracy (95% CI). The table compares *Initial* (baseline), *Optimized* (RECAP-PATH), and *In-context Learning* strategies across binary (N/IC, DCIS/IC) and multiclass (N/DCIS/IC) tasks. The final row, *Cross-Model* (N/IC), evaluates prompt transferability by optimizing diagnostic criteria on the peer backbone (e.g., using GPT-4o-derived prompts for Gemini 2.0 Flash inference).

guided optimization increased clinical coherence ratings of image descriptions by nearly 20%, demonstrating the importance of domain expertise in bridging the gap between algorithmic inference and medical practice.

A central advancement of our framework is its ability to provide case-specific transparency through the MLLM's image-level descriptions, which explicitly articulate the visual features underlying each diagnostic decision. As shown in Fig. 3C, for a representative normal tissue sample, the model identifies hallmark features of benign histology, including well-organized tubular architecture, uniform glandular cell size, absence of prominent nucleoli, and abundant cytoplasm. These findings are fully consistent with expert pathological criteria for this image. For invasive carcinoma, the descriptions highlight malignant features such as infiltrative growth patterns, nuclear pleomorphism, prominent nucleoli, and stromal invasion. These outputs parallel expert annotations, showing that the model justifies its predictions in a clinically intelligible way. We reviewed misclassified cases and found that most errors arose from a limited field of view or imaging artifacts that obscured key morphologic features (Fig. S7). These failures indicate the framework's sensitivity to input image quality and suggest that artifact detection or correction could further improve robustness. Overall, our method delivers pathology-aligned explanations that strengthen the diagnostic trustworthiness.

### RECAP-PATH enables interpretable subtype classification in breast cancer pathology

Beyond distinguishing normal from malignant cases, precise differentiation between breast cancer subtypes, particularly ductal carcinoma in situ (DCIS) and invasive carcinoma (IC), is essential for accurate prognosis and treatment planning. DCIS remains confined to the ductal system, is typically associated with an excellent prognosis, and is often managed conservatively. In contrast, IC is defined by stromal invasion, carries a higher risk of metastasis, and generally requires more aggressive intervention. As shown in Fig. 4A, RECAP-PATH achieved strong classification performance, with true positive rates of 0.85 for DCIS and 0.90 for IC. Through prompt refinement, the MLLM autonomously generated subtype-specific diagnostic criteria that emphasized key distinguishing features (Fig. 4B), including confinement of malignant cells within ducts, the presence or absence of stromal invasion, and contrasts between intraductal and infiltrative growth patterns. These features are consistent with canonical DCIS presentations, as illustrated in Fig. 4C. Notably, these diagnostic guidelines emerged spontaneously without manual annotation of subtype-specific cues, indicating the model's capacity to internalize and articulate clinically relevant histopathologic distinctions. Consistent with our earlier findings, manually curated clinical prompts underperformed in these complex tasks, achieving 79.5% accuracy for DCIS/IC and 60.5% for the multiclass setting (Table 1, Fig. S2). This reinforces that RECAP-PATH's automated

optimization effectively bridges the gap between generic medical knowledge and model-specific reasoning capabilities.

Furthermore, the UMAP visualization of MLLM-generated descriptions in Fig. 4D revealed two separated clusters corresponding to DCIS and IC, indicating that the model's descriptions were both subtype-specific and semantically disentangled. Critically, the MLLM-generated descriptions for DCIS consistently highlighted hallmark histopathologic features, such as marked nuclear pleomorphism and confinement of atypical cells within intact ducts, closely mirroring established diagnostic criteria. This semantic separation confirms the model's ability to generate phenotype-specific explanations and articulate true morphologic differences between subtypes.

We further extended the classification task to a more complex multi-class setting involving normal tissue, DCIS, and IC. Multiclass pathology triage is inherently challenging due to diagnostic ambiguity and overlapping morphological features. Despite these complexities, the model maintained strong performance, as demonstrated by the confusion matrices in Fig. 4E and UMAP visualizations in Fig. 4F. The semantic embeddings of descriptions for DCIS and IC formed well-separated clusters, underscoring the model's capacity to distinguish invasive from non-invasive disease. A subset of normal cases was misclassified as DCIS, likely reflecting shared architectural features such as ductal confinement. These errors suggest that while the framework effectively captures global tissue architecture, further refinement to emphasize subtler cytological cues, such as nuclear pleomorphism, could reduce residual ambiguity. Overall, these results support RECAP-PATH's ability to scale from binary to multiclass diagnostic reasoning while also defining opportunities for future enhancement.

We then assessed robustness and model-agnostic versatility. We ran experiments with two independent state-of-the-art general-purpose MLLMs, Google Gemini 2.0 Flash[51] and OpenAI gpt-4o-2024-05-13[60], using black-box API access and no finetuning. As summarized in Table 1, the optimized prompt consistently outperformed the initial prompt across all experimental settings. Because these models are general-purpose rather than pathology-specific, the cross-model gains indicate that our prompt optimization strategy transfers across foundation models without white-box access, supporting robustness to model choice and practical deployment in API-constrained clinical environments. In Table 1, the 95% confidence intervals define the performance margins of the framework. These intervals quantify the variance inherent in the prompt generation process, establishing the lower and upper bounds of diagnostic accuracy to demonstrate the method's stability across variations in prompt phrasing.

We also assessed prompt transferability. We applied prompts optimized by GPT-4o to Gemini 2.0 Flash for inference, and vice versa (Table 1). This cross-model application yielded substantial accuracy gains over unoptimized baselines; for instance, applying GPT-4o-optimized criteria to

(A)

Expert knowledge augmentation

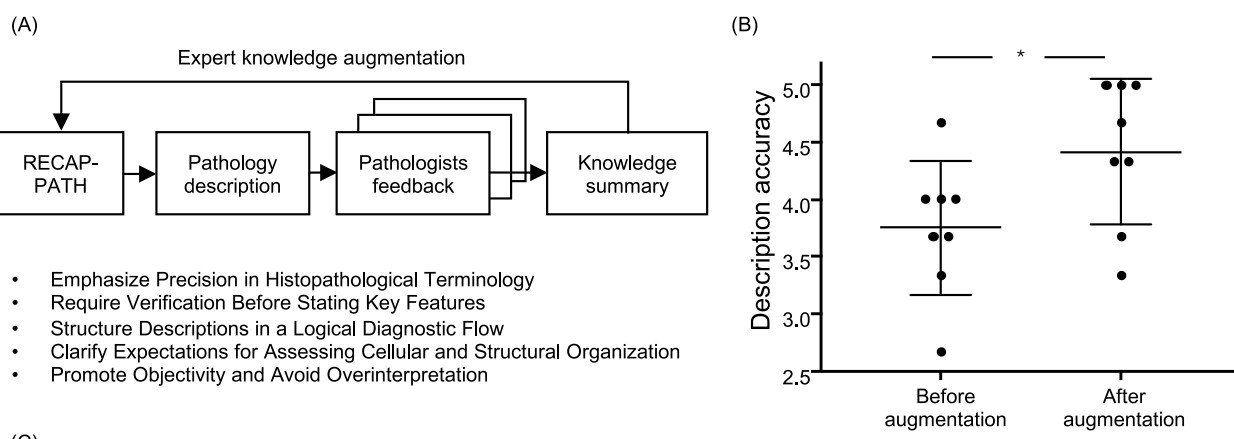

- Emphasize Precision in Histopathological Terminology
- Require Verification Before Stating Key Features
- Structure Descriptions in a Logical Diagnostic Flow
- Clarify Expectations for Assessing Cellular and Structural Organization
- Promote Objectivity and Avoid Overinterpretation

(C)

Normal

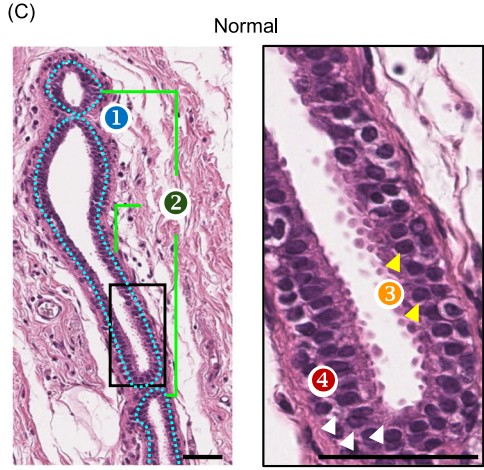

**RECAP-PATH description: normal**

o  Tissue Organization: **The tissue architecture is well-organized, displaying regular and distinct tubular structures**❶...

o  Cellular Features: **The cells lining the tubules are uniform in size and shape**❷...

o  Nuclear Morphology: **The nuclei are round, with finely dispersed chromatin and no prominent nucleoli**❸. There is no evidence of hyperchromasia, pleomorphism, or increased nuclear-to-cytoplasmic ratio.

o  Cytoplasmic Characteristics: **The cytoplasm is abundant and eosinophilic**❹. There are no atypical features in the cytoplasm.

o  Diagnostic Markers: There are no features suggestive of atypical ductal hyperplasia … The overall appearance of the tissue is consistent with normal breast ductal epithelium.

Invasive carcinoma

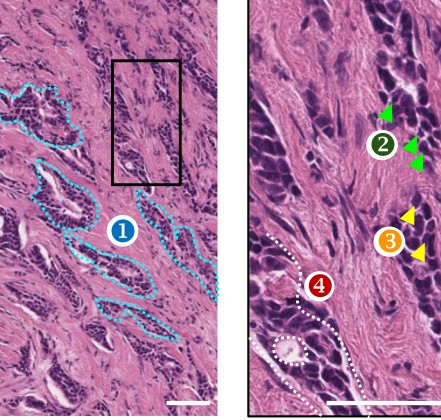

**RECAP-PATH description: invasive carcinoma**

o  Tissue Organization: **The tissue architecture is significantly disrupted, with a loss of normal breast tissue organization. Abnormal, irregular, and infiltrative growth patterns**❶...

o  Cellular Features: **Cellular pleomorphism is prominent, with marked variation in size and shape**❷ of the tumor cells. The cells exhibit a high nuclear-to-cytoplasmic ratio ...

o  Nuclear Morphology: **Nuclei are hyperchromatic, ... Prominent nucleoli**❸ ...

o  Cytoplasmic Characteristics: The cytoplasm is … scant and eosinophilic .

o  Diagnostic Markers: The combination of **stromal invasion, cellular pleomorphism**❹, hyperchromatic nuclei, prominent nucleoli, and irregular growth patterns ...

**Fig. 3 | Pathologist knowledge augmentation for RECAP-PATH optimization. A** Integration of expert feedback into the RECAP-PATH framework. Three board-certified pathologists provided blinded evaluations of LLM-generated image descriptions across normal and invasive carcinoma cases, rating their precision and histopathological accuracy. These assessments were incorporated into the refinement process. The assessment summary is shown. **B** Pathologist ratings demonstrated that incorporating expert feedback improved the clinical coherence and histopathological correctness of generated descriptions by nearly 20%. Data represent $n = 9$ independent expert evaluations (3 evaluations performed by each of three board-certified pathologists), with error bars indicating the standard deviation. * indicates statistical significance ($p < 0.05$, two-sided Student's $t$-test). **C** Case-level analysis of optimized outputs. Representative examples of image-specific diagnostic narratives illustrate how the optimized diagnosis criteria guide the MLLM to identify key histopathological features. For benign samples, the model describes hallmark features such as organized tubular architecture, uniform gland size, absence of nucleoli, and abundant cytoplasm. For invasive carcinoma, it highlights malignant patterns including infiltrative growth, nuclear pleomorphism, prominent nucleoli, and stromal invasion. Scale bars = 50 μm.

Gemini 2.0 Flash achieved 86.8% accuracy (compared to ~71% baseline). While the highest performance was consistently achieved when the optimization and inference models were identical, these results confirm that the generated diagnostic logic captures transferable morphological principles rather than model-specific artifacts.

Lastly, to address clinical data privacy requirements, we validated the framework on the open-source Qwen2.5-VL-72B-Instruct model. Focusing on binary classification (Normal vs. Invasive Carcinoma) as a proof-of-concept due to excessive local computational turnaround times, RECAP-PATH improved diagnostic accuracy from 55.5% (95% CI: 49.8–61.2%) to

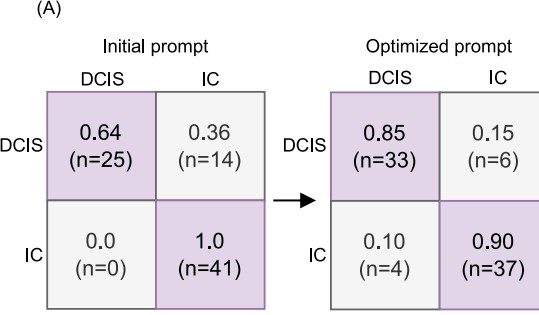

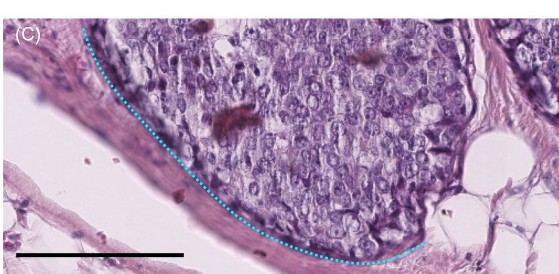

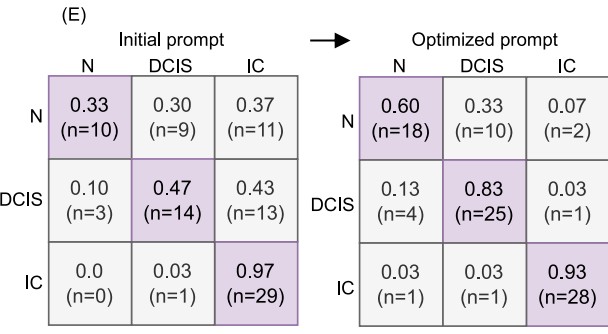

RECAP-PATH description: DCIS

**Overall Architecture:** … appear to be mostly contained within a duct-like structure... The cells exhibit significant pleomorphism, with varying sizes and shapes...
**Reactive Stroma: Absent.** The stroma around the tumor does not exhibit a clear desmoplastic reaction.
**Intraductal Location: Present.** A large portion of the cells appears to remain within the ductal structure.

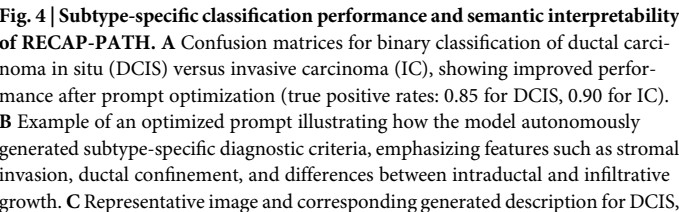

(B) Optimized prompt (diagnosis criteria)

… analyze the breast tumor pathology slide, differentiating between … DCIS and Invasive Carcinoma based exclusively on its morphology.
1. **Distinguishing Feature**: The critical difference hinges on whether the neoplastic cells are restricted to the confines of the mammary ducts (DCIS) or have breached the ductal basement membrane and infiltrated the surrounding stroma (Invasive Carcinoma).
2. **Invasion Indicators**: Stromal Extension: … irregular clusters or individual cells extending into the stroma beyond the ductal architecture...**Reactive Stroma**: … desmoplastic response. **Myoepithelial Cell Absence**: … assess the presence or absence of the myoepithelial layer around the duct …
3. **DCIS Indicators**: **Intraductal Location**: The neoplastic cells are **contained within the ducts** …the overall confinement is paramount. Circumscription: The tumor mass often exhibits relatively well-defined borders... Microinvasion Awareness: Remain cognizant that DCIS may exhibit minimal microinvasion…

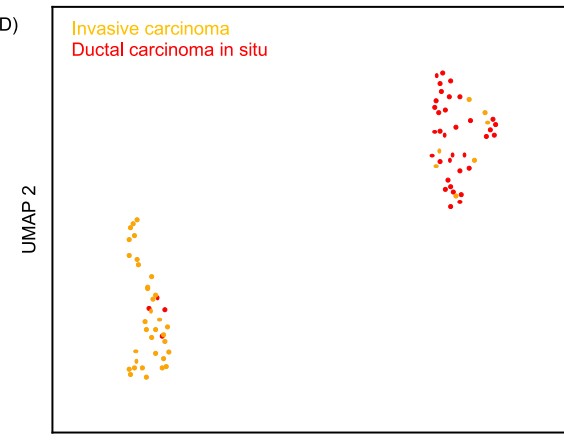

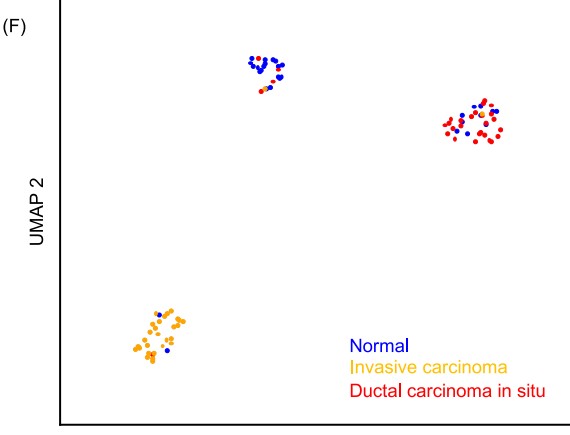

**Fig. 4 | Subtype-specific classification performance and semantic interpretability of RECAP-PATH. A** Confusion matrices for binary classification of ductal carcinoma in situ (DCIS) versus invasive carcinoma (IC), showing improved performance after prompt optimization (true positive rates: 0.85 for DCIS, 0.90 for IC). **B** Example of an optimized prompt illustrating how the model autonomously generated subtype-specific diagnostic criteria, emphasizing features such as stromal invasion, ductal confinement, and differences between intraductal and infiltrative growth. **C** Representative image and corresponding generated description for DCIS, highlighting hallmark features aligned with established pathological criteria. Scale bar = 100 μm. **D** UMAP visualization of description embeddings, demonstrating clear semantic separation between DCIS and IC, consistent with phenotype-specific explanations. **E** Confusion matrices for multiclass classification (N, DCIS, IC), showing progressive performance gains after optimization. **F** UMAP visualization of description embeddings in the multiclass setting, demonstrating improved clustering and subtype differentiation, with most errors involving normal cases misclassified as DCIS due to overlapping ductal features.

67.0% (95% CI: 62.5–71.5%), demonstrating its compatibility with locally deployable, privacy-preserving workflows.

### RECAP-PATH generalizes across datasets: prostate Gleason grading and breast histology

Complementing our cross-model analysis, we next evaluate the adaptability of our framework to additional histopathology datasets. We first applied the framework to the Breast Cancer Histology (BACH) dataset[49], focusing on distinguishing normal breast tissue from invasive carcinoma for direct comparison with our previous BRACS-based results. Unlike BRACS, the BACH dataset is smaller in size and has lower spatial resolution (~0.42 μm/pixel, approximately half that of BRACS). Despite these differences, our prompt optimization pipeline achieved a classification accuracy comparable to that observed with BRACS (Fig. 5A), following a similar diversification and optimization trajectory (Fig. 5B). This consistency suggests that our prompt-learning mechanism generalizes effectively across datasets. As in our prior analysis, the optimized prompts highlighted comprehensive morphological descriptors, explicitly incorporating architectural

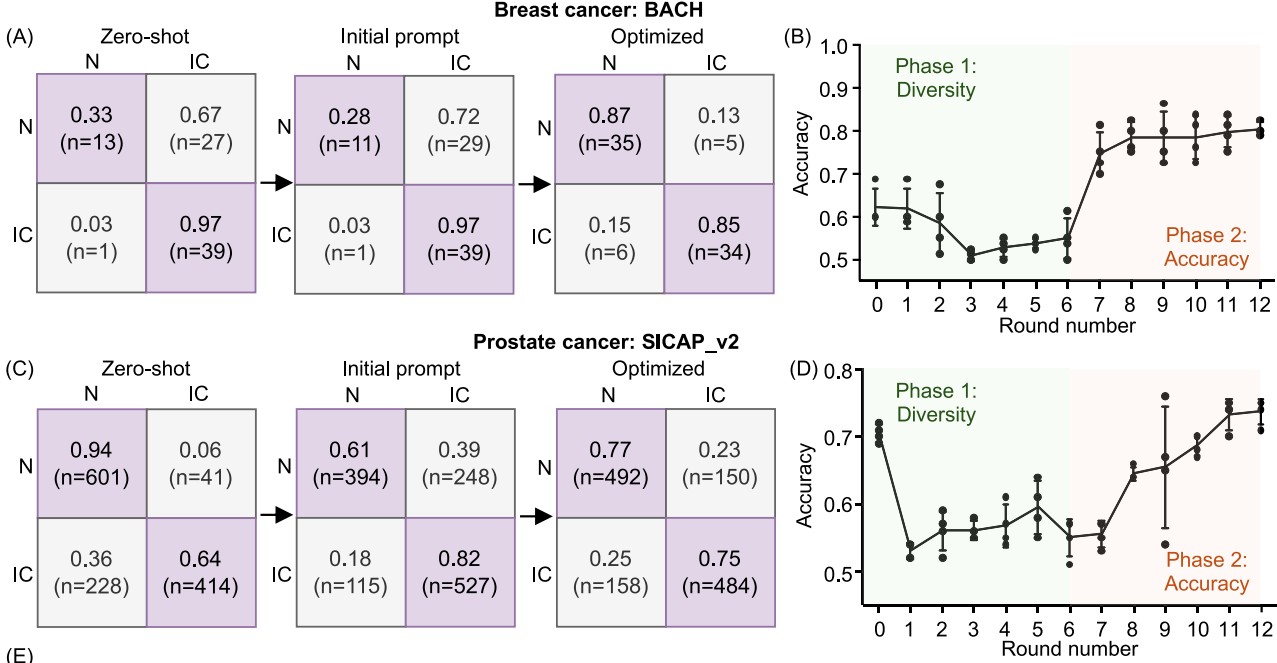

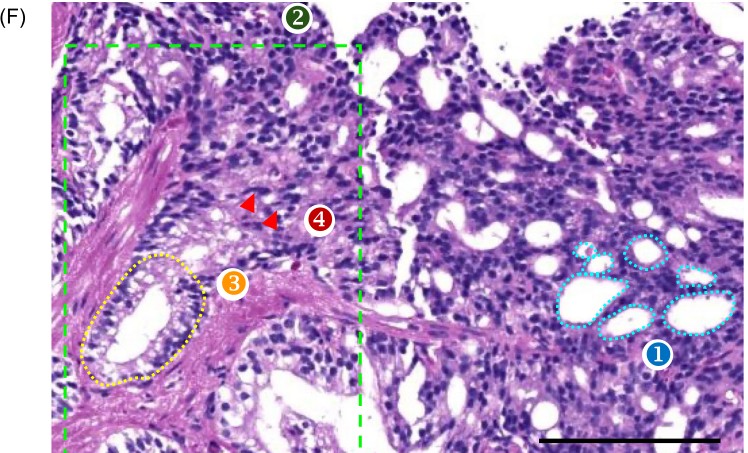

**Fig. 5 | Adaptability of RECAP-PATH across pathology datasets. A** Confusion matrices for normal versus invasive carcinoma in the BACH dataset, showing improved performance after prompt optimization despite lower resolution and smaller dataset size compared to BRACS. **B** Accuracy trajectory in BACH, demonstrating a similar non-monotonic two-phase learning dynamic as observed in BRACS. **C** Confusion matrices for benign versus malignant classification in the prostate cancer SICAPv2 dataset, showing balanced performance gains after optimization. **D** Reproduction of the two-phase learning dynamics in SICAPv2. **E** Example optimized diagnostic criteria in prostate histology, illustrating key features such as glandular architecture, arrangement, and cytological atypia. **F** Case-level description of a malignant prostate sample (Gleason 4 + 4), where the optimized prompt guided the model to identify hallmark features, including cribriform architecture, infiltrative growth, and nuclear pleomorphism, consistent with expert diagnostic criteria. Scale bar = 100μm. Error bars in (**B**) and **D** indicate standard deviation derived from $n = 4$ independent optimization experiments.

organization, the presence and morphology of myoepithelial cells, cellular density, and stromal composition. These refinements facilitated more precise discrimination between normal breast tissue and invasive carcinoma.

We also examined a substantially different dataset consisting of prostate cancer images annotated with patch-level Gleason grades (SICAPv2)[50]. Despite differences in tissue type and diagnostic standards, RECAP-PATH achieved high classification accuracy after prompt learning, generating balanced predictions (Fig. 5C) and clinically reasonable analyses. The two-phase learning dynamics was also reproduced in this dataset (Fig. 5D). The optimized prompts were adapted to prostate histology, guiding the model toward critical diagnostic hallmarks, including overall glandular architecture, glandular arrangement, and cytological atypia (Fig. 5E). The prompts further directed the model to provide Gleason grading and justification based on these criteria (Fig. 5F). This methodological approach enabled the MLLM to produce structured, clinically interpretable outputs aligned with routine diagnostic workflows. For instance, the model correctly identified hallmark features such as cribriform architecture, infiltrative glandular growth, and nuclear pleomorphism in a case diagnosed as malignant with a Gleason score of $4 + 4$, consistent with the ground-truth annotation. Collectively, these experiments demonstrate the dataset-specific adaptability of RECAP-PATH. By iteratively refining prompts to reflect domain-specific histological features, our framework not only enhances classification accuracy but also produces pathologically coherent reasoning across diverse diagnostic contexts.

## Discussion

We introduce RECAP-PATH, a two-phase learning framework for trustworthy AI pathology. It achieves >0.9 accuracy in distinguishing normal from invasive carcinoma and delivers up to ~20% gains across tasks and models, while providing pathologist-aligned rationales. By generating image-grounded explanations[34], RECAP-PATH highlights the promise of MLLMs and directly addresses the opacity of post-hoc methods in conventional AI pathology. Although general-purpose MLLMs excel at image description, their pathology performance often lags without adaptation. Prior efforts, such as in-context examples[33] and domain-specific fine-tuning[30], enhance alignment but require extensive high-quality data and often fall short of delivering transparent reasoning.

Parallel to these adaptation strategies, recent work has formalized prompt engineering into an automated optimization problem. Search-based methods, most notably APE[54], Optimization by PROmpting (OPRO)[45], and PromptAgent[61], leverage LLMs to iteratively generate and select instructions. However, our supplementary benchmarking suggests that broad search strategies like APE yielded slightly lower classification accuracy in this context compared to our framework. Distinct from search, feedback-driven frameworks like Automatic Prompt Optimization (APO)[44], TextGrad[62], and Reflexion[63] mimic gradient descent by using textual critiques to refine prompts. While effective, their primary focus on maximizing scalar accuracy metrics risks encouraging shortcut learning, converging on concise cues that optimize scores, potentially at the expense of the step-by-step reasoning necessary for clinical auditability. RECAP-PATH diverges from these approaches with a self-learning, model-agnostic framework that prioritizes a diversify-to-optimize axis. Rather than optimizing solely for the final label, it aims to address this balance by refining diagnosis criteria to yield evidence-linked image descriptions, ensuring the model articulates morphological features before prediction.

From a biomedical perspective, this description-first approach prioritizes practical auditability over intrinsic model interpretability. By producing explicit, morphology-level narratives that document the evidence behind each call, it generates the traceable outputs that clinicians and tumor boards increasingly expect for AI in health. Its description-first pipeline elicits pathology-relevant criteria (invasion, architecture, nuclear features, stroma, mitoses) and integrates them with the image for final decisions, aligning the model's reasoning with established diagnostic practice and avoiding the unstable cues of post-hoc explainer[64]. Importantly, blinded expert-in-the-loop evaluation shows that pathologist feedback measurably improves the histopathological correctness of histology image descriptions[65]. These elements position RECAP-PATH as a pragmatic, evidence-focused solution that supports clinical trust and responsible deployment in pathology.

A further limitation inherent to generative models is the risk of hallucinating morphological features. While our optimization process anchors descriptions to valid diagnostic criteria to mitigate unsupported claims, the generated rationales are intended to serve as a tool for expert verification and error analysis rather than as an infallible window into the model's internal decision-making. Another limitation is that the optimized prompt remains dataset-specific: each new dataset requires a separate optimization process, as prompts tuned on one cohort (e.g., a particular breast cancer dataset) may not generalize effectively to another, even within the same disease type. Future studies should investigate slot-based calibration with a small number of labeled examples as a lightweight alternative to re-running prompt optimization from scratch, enabling more efficient cross-dataset generalization. Lastly, our experiments relied on ROI/patch crops rather than whole-slide images (WSIs), sidestepping tissue selection and aggregation steps that are critical for clinical realism. An immediate next step is to embed RECAP-PATH in a full WSI pipeline and benchmark against external datasets and state-of-the-art slide-level baselines[66–68]. Finally, the framework depends on vendor-hosted MLLMs, raising concerns about undocumented updates and version drift that may undermine reproducibility[69]. To address this, future work should stress-test open-source local models for comparability and adopt standardized, multi-metric evaluation protocols and clinical reporting guidelines to ensure robustness, calibration, and transparent documentation[70–72].

In sum, RECAP-PATH shows that reasoning itself can be optimized into a reliable, auditable signal—transforming MLLMs from opaque classifiers into systems capable of delivering clinically aligned decisions. By embedding domain-specific diagnostic criteria directly into the decision process, it bridges methodological innovation in machine learning with the practical demands of pathology. More broadly, it exemplifies a paradigm in which interpretability and performance are not opposing goals but mutually reinforcing, charting a path toward AI systems that are both scientifically rigorous and ready for real-world clinical integration.

## Data availability

Source data for Figures 2A-C, 3B, 5B, and 5D are provided in the Supplementary Data. Additionally, Python code with embedded raw input data for reproducing Figures 2F, 4D, and 4F is available in the Supplementary Data. The datasets used in this study are publicly available and can be downloaded from https://www.bracs.icar.cnr.it/(BRACS), https://iciar2018-challenge.grand-challenge.org/Dataset/(BACH), and https://data.mendeley.com/datasets/9xxm58dvs3/1(SICAPv2).

## Code availability

The code for this work is publicly available at GitHub and Zenodo[73].

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

## Acknowledgements

This work was partially supported by NSF (CBET-2244760, DBI-2325121, IIS-2048280) and NIH NIGMS (R35GM146735).

## Author contributions

Y.H., C.J.H., and N.Y.C.L. conceived the project. Y.H. and L.E. developed the framework. Y.H., K.-C.K., and L.E. performed the tests. N.T.L., C.Y.H., and A.O.K. conducted the blind evaluation and developed the expert knowledge augmentation. All authors contributed to data analysis, paper writing, and review.

## Competing interests

The authors declare no competing interests.
