## [Transparent Peer Review file · Communications Medicine]

Adaptive Diagnostic Reasoning Framework for Pathology with Multimodal Large Language Models

Corresponding Author: Professor Neil Lin

Version 0:

Reviewer comments:

Reviewer #1

(Remarks to the Author)

Summary:

The manuscript "Adaptive Diagnostic Reasoning Framework for Pathology with Multimodal Large Language Models" proposes an iterative approach to refine prompts for multimodal large language models (MLLMs) without finetuning the MLLM. Through an iterative "thinking process" which is aligned to pathologist's explanations and designed to achieve more interpretable AI-based decisions, prompts are generated that improve the classification performance on three downstream datasets for cancer type classification. The manuscript is well-written, nicely organized, and has a good presentation quality. While interpretability and the black-box nature of neural networks are important topics for clinical implementations of AI algorithms and the manuscript presents an interesting approach to improve this, I have the following concerns:

Major issues:

- In general, having an AI-based written form of reasoning is not the same as model interpretability. In fact, the model also hallucinates morphological features as seen in the pathologist evaluation. The manuscript does not address this sufficiently.
- To assess the approach, reference for the performance are necessary. This should be provided with respect to the proposed method, for example by adding a simple in-context learning baseline providing the clinical definitions and morphological features necessary for the task at hand. And further, to better assess the potential margins of this prompt tuning, lower and upper bars of the performance should be given by random chance (or an LLM baseline without image input or better a baseline with LLM-generated morphology descriptions of the image) and the supervised performance of classifiers on these tasks. Without these, claims of performance improvement are difficult to assess.
- It is not clear to me why the notion of generalization is used, given that the pipeline is not fitted to a specific dataset. It would however be interesting to see how well the generated prompts generalize across samples and across models. If these generalize well, the optimized prompts could be made available for use in this context.

Minor issues and presentation

- Figures 2D, 4E, 5A, 5C: Add axis labels to the confusion matrices.
- Table 1: Consider merging accuracy and its confidence interval into a single line: Acc (lower CI, upper CI).
- Typos / Wording: Page 1: "Bu -> But"

I enjoyed reading the paper and it presents an interesting approach. I would recommend to address the issues that I have raised above to improve the value of the manuscript for the scientific community and to substantiate the key claim regarding interpretability and clinical utility.

Reviewer #2

(Remarks to the Author)

The paper presents RECAP-PATH, a framework for interpretable diagnostic reasoning in digital pathology using a two-phase prompt optimization strategy with multimodal large language models (MLLMs). The method enables pathology-style description generation using general-purpose models without finetuning, showing strong accuracy on datasets such as BACH and SICAP. The study is well evaluated, with convincing results and validation by board-certified pathologists.

A. Strengths:

1. Novel prompt optimization strategy: The two-phase optimization (diversification + refinement) is well-motivated and effectively balances reasoning diversity and diagnostic accuracy. It is conceptually elegant and clearly presented.
2. Experiment design: The experiments are well structured and anticipate the reader's questions, showing strong experimental intuition.
3. Expert evaluation: The inclusion of three board-certified pathologists and a human-in-the-loop setup adds valuable external validation and clinical credibility.

B. Weaknesses:

B.1. Major Concerns

1. Prompt optimization baselines: Compare RECAP-PATH against other prompt optimization methods to demonstrate its advantages.
2. Open-source models: Since data privacy is critical in the medical domain, it would be valuable to test RECAP-PATH with open-source models that can be run locally (e.g. MedGemma, Qwen-VL).
3. Related work: Expand the discussion of prior prompt optimization strategies to provide clearer context for the proposed approach.

B.2. Minor / Clarification Points

1. Figure 2: The gray samples are unclear at first sight. Improve the visualization or caption for better readability.

B.3. Optional

1. Baselines: Include a comparison to multimodal pathology models such as Quilt-LLaVA to better contextualize performance.

C. Grammar/Typos

1. Typo: "Image description generation and diganosis" (p. 11).
2. Writing error: "To assess the clinical relevance and interpretability of RECAP-PATH, we conducted a blind evaluation survey involving three board-certified pathologists. with images and their corresponding descriptions generated by MLLM."

Version 1:

Reviewer comments:

Reviewer #1

(Remarks to the Author)

My concerns were thoroughly addressed by the authors and implemented in the manuscript. I have no further comments and would recommend the manuscript for publication in the current shape.

Reviewer #2

(Remarks to the Author)

Thank you for the revision and detailed responses. I appreciate the refined interpretability framing and the addition of the CLIP-based baseline. The added baselines, validation on an open-source MLLM, and improved contextualization within the prompt optimization literature strengthen the manuscript and clearly address the points raised during the review process. I have no further concerns.

Reviewer # 1

Summary:

The manuscript “Adaptive Diagnostic Reasoning Framework for Pathology with Multimodal Large Language Models” proposes an iterative approach to refine prompts for multimodal large language models (MLLMs) without finetuning the MLLM. Through an iterative “thinking process” which is aligned to pathologist’s explanations and designed to achieve more interpretable AI-based decisions, prompts are generated that improve the classification performance on three downstream datasets for cancer type classification. The manuscript is well-written, nicely organized, and has a good presentation quality. While interpretability and the black-box nature of neural networks are important topics for clinical implementations of AI algorithms and the manuscripts presents an interesting approach to improve this, I have the following concerns:

Response:

We thank the reviewer for the constructive feedback and for recognizing the manuscript’s organization and the potential of our iterative prompting approach to improve MLLM performance. We have carefully addressed the reviewer’s concerns regarding the distinction between AI-generated reasoning and intrinsic interpretability. Accordingly, we have refined our terminology throughout the text to emphasize “audit-ready” diagnostics rather than intrinsic transparency, while explicitly addressing the limitations concerning feature hallucination. To provide a rigorous assessment of performance margins, we have incorporated the requested benchmarks, including a supervised CLIP-based “upper bar” and manual in-context learning baselines. Finally, we clarified our usage of “generalization” versus “adaptability” and included new experiments demonstrating cross-model prompt transferability (e.g., GPT-4o optimized prompts deployed on Gemini), confirming that the diagnostic logic discovered by our framework captures robust, model-agnostic morphological principles.

Major issue 1:

In general, having an AI-based written form of reasoning is not the same as model interpretability. In fact, the model also hallucinates morphological features as seen in the pathologist evaluation. The manuscript does not address this sufficiently.

Response:

We appreciate the reviewer highlighting this critical distinction and agree that AI-generated text differs from intrinsic model interpretability, particularly given the risk of hallucinated morphological features. Accordingly, we have revised the manuscript to clarify that our objective is not interpretability in the traditional sense, but rather the generation of pathologist-readable, audit-ready rationales designed to facilitate human-in-the-loop verification. Specifically, we have updated the terminology in the Abstract, Introduction, and Results section headings to replace broad claims of ‘interpretability’ with ‘audit-ready diagnostics’ and ‘rationale-based triage.’ Furthermore, we have expanded the Discussion to transparently address the limitations regarding feature hallucination, emphasizing that while our automatic prompt optimization systematically reduces unsupported claims by aligning outputs with diagnostic criteria, these rationales are intended to serve as a tool for clinical auditability and error analysis rather than as an infallible window into the model’s internal decision-making.

Revisions:

1. Page 1, Abstract Rationale:

We have replaced the word “interpretable” with “audit-ready”.

2. Page 3, Introduction, Paragraph 5 Rationale:

Original: "...we propose RECAP-PATH (REasoning and Classification via Automated Prompting in PATHology images), an **interpretable** framework that leverages MLLMs..."

Revised: "...we propose RECAP-PATH (REasoning and Classification via Automated Prompting in PATHology images), a framework **designed to facilitate clinical verification** that leverages MLLMs..."

3. Page 3, Results Heading Rationale:

Original: "Two-Phase Prompt Optimization Enables Accurate and **Interpretable** Triage of Invasive Carcinoma"

Revised: "Two-Phase Prompt Optimization Enables Accurate and **Rationale-Based** Triage of Invasive Carcinoma"

4. Page 10, Discussion, Paragraph 2 Rationale:

Original: "From a biomedical perspective, RECAP-PATH's principal advance is practical auditability: by producing explicit, morphology-level narratives..."

Revised: "From a biomedical perspective, this description-first approach **prioritizes practical auditability over intrinsic model interpretability**. By producing explicit ..."

5. Page 10, Discussion, Paragraph 3 (Limitations) Rationale:

Original: "One limitation is that the optimized prompt remains dataset-specific: each new dataset requires a separate optimization process..."

Revised: "A further limitation inherent to generative models is the risk of hallucinating morphological features. While our optimization process anchors descriptions to valid diagnostic criteria to mitigate unsupported claims, the generated rationales are intended to serve as a tool for expert verification and error analysis rather than as an infallible window into the model's internal decision-making. Another limitation is that the optimized prompt remains dataset-specific..."

Major issue 2:

To assess the approach, references for the performance are necessary. This should be provided with respect to the proposed method, for example by adding a simple in-context learning baseline providing the clinical definitions and morphological features necessary for the task at hand. And further, to better assess the potential margins of this prompt tuning, lower and upper bars of the performance should be given by random chance (or an LLM baseline without image input or better a baseline with LLM-generated morphology descriptions of the image) and the supervised performance of classifiers on these tasks. Without these, claims of performance improvement are difficult to assess.

Response:

We thank the reviewer for these constructive suggestions regarding performance benchmarking. We agree that establishing clear lower and upper performance bounds is essential for assessing the validity of our prompt optimization approach. We have addressed these points through the following revisions:

1. *In-Context Learning Baseline:* As suggested, we have implemented an in-context learning baseline that incorporates manually curated clinical definitions and morphological features for the N/IC, DCIS/IC, and N/DCIS/IC classification tasks. Our analysis reveals that manually adding context does not yield consistent performance gains, as effectiveness relies heavily on the specific phrasing and the MLLM's ability to utilize the context. In contrast, RECAP-PATH provides a systematic, automated optimization that consistently aligns the context with the model's capabilities to improve performance.

2. *Supervised "Upper Bar" Comparison:* We established an "upper bar" for performance. We compared RECAP-PATH against a supervised linear classifier trained on CLIP (Contrastive Language-Image Pre-training) embeddings. As noted in the revised manuscript, this supervised classifier achieves ~92.5% accuracy on N/IC classification. RECAP-PATH (using Gemini 2.0 Flash) achieves comparable accuracy. However, RECAP-PATH operates under stricter constraints: no weight updates and no white-box access. Crucially, it provides pathologist-readable rationales, which the supervised classifier cannot do.

3. *Performance Margins and Confidence Intervals:* We believe that the confidence intervals provided in the original submission already address performance margins by capturing the distribution of accuracy resulting from prompt phrasing variations. These intervals quantify prompt-induced variance, effectively establishing the stability and performance bounds of the optimization process. We have revised the manuscript to clarify this interpretation and explicitly defined the random chance baseline in the figure legends.

4. *Clarification on the No-Image Baseline:* We would appreciate further guidance on the "LLM baseline without image input." We interpreted this as classifying cases using only pre-generated morphological descriptions. However, the UMAP plots in Fig. 2F show a dominant correlation between the generated description and the classification decision. This is due to the strong semantic cues already embedded in the descriptions. Therefore, removing the image did not provide independent insights into the framework's mechanism. If we misunderstood the intent of this specific baseline, we are happy to conduct additional experiments.

Revisions:

1. A paragraph describing the CLIP results has been added on Page 3, at the end of the paragraph discussing Figure 2A (learning dynamics for Normal vs. Invasive Carcinoma), specifically after the sentence ending in "...clinically audit-ready." The added paragraph reads:
"To benchmark this performance, we established a supervised upper bar using a linear classifier trained on Contrastive Language-Image Pre-training (CLIP) embeddings ~\cite{radford2021learning}, which achieved 92.5\% accuracy. ... RECAP-PATH attains comparable results without requiring weight updates or white-box access, while providing pathologist-readable rationales ..."
2. In the same paragraph above, we added the following discussion:
"Additionally, an in-context learning baseline using manually curated clinical definitions yielded lower performance (86.0\% for N/IC) than RECAP-PATH (Table 1, Fig. S2).... These comparisons indicate that neither static expert knowledge nor general-purpose instruction tuning is sufficient without the systematic, reasoning-driven alignment provided by our optimization."
3. The following discussion is included on Page 7, in the subsection "RECAP-PATH enables interpretable subtype classification...", at the end of the paragraph discussing the multiclass

performance:

"Consistent with our earlier findings, manually curated clinical prompts underperformed in these complex tasks, achieving 79.5% accuracy for DCIS/IC and 60.5% for the multiclass setting (Table 1, Fig. S2). This reinforces that RECAP-PATH's automated optimization effectively bridges the gap between generic medical knowledge and model-specific reasoning capabilities."

4. We have added a paragraph in the "Comparison methods and validation benchmarks" subsection of the Methods describing our use of CLIP as a baseline.
"\item \textbf{CLIP} \cite{radford2021learning} (Contrastive Language–Image Pretraining) serves as a fundamental vision-language baseline. It is trained by jointly learning image and text encoders to map both modalities into a shared embedding space using a contrastive objective. The model optimizes a symmetric cross-entropy~\cite{oord2018representation} (InfoNCE) loss over batches of image-caption pairs to maximize the cosine similarity of matched pairs while minimizing it for non-matching ones. We utilized CLIP for its zero-shot classification capabilities, where the image embedding is compared directly to the embeddings of class-specific text descriptions without additional training."
5. We inserted the following discussion in the subsection "RECAP-PATH enables interpretable subtype classification in breast cancer pathology" of the Results section:
"... In Table 1, the 95% confidence intervals define the performance margins of the framework. These intervals quantify the variance inherent in the prompt generation process, establishing the lower and upper bounds of diagnostic accuracy to demonstrate the method's stability across variations in prompt phrasing."
6. The in-context learning results below have been added to Table 1. The used prompts have been included in SI as Fig. S2. We have also appended them below for the Reviewer's convenience.

Categories	Model	Acc	Std	Upper	Lower
N/IC	gemini 2.0 001 flash	0.8600	0.0094	0.8730	0.8470
DCIS/IC	gemini 2.0 001 flash	0.7950	0.0100	0.8089	0.7811
N/DCIS/IC	gemini 2.0 001 flash	0.6047	0.0086	0.6166	0.5928

N/IC prompt.

You are assisting with a breast cancer pathology image classification task. The goal is to distinguish normal breast tissue from invasive breast carcinoma in H&E-stained microscopy images. Below is the clinical and morphological context you must rely on.

Normal Breast Tissue:

- Normal breast tissue is composed of terminal duct–lobular units (TDLUs), which contain small acini arranged into lobules.
- Ducts and acini have a dual cell layer: an inner luminal epithelial layer and an outer myoepithelial layer. The myoepithelial layer is a key indicator of non-invasive epithelium.
- Surrounding stroma contains a mixture of fibrous connective tissue and adipose tissue.
- Morphological features include well-organized architecture, intact myoepithelial layers, uniform small nuclei with low atypia, low mitotic activity, and the absence of desmoplastic stromal reaction.

Invasive Breast Carcinoma:

- Invasive breast carcinoma is defined clinically as malignant epithelial cells that have breached the basement membrane and infiltrated the surrounding stroma.
- This distinction directly affects cancer staging, prognosis, and treatment decisions.
- Key morphological features of invasive carcinoma include: loss of the normal duct/lobular architecture; absence of the myoepithelial cell layer around tumor structures; infiltrative patterns such as irregular glands, nests, cords, or single tumor cells invading stroma; marked cytologic atypia such as enlarged pleomorphic nuclei and prominent nucleoli; increased mitotic figures; and the presence of desmoplastic (fibrotic) reactive stroma.
- Common invasive patterns include malformed ducts with irregular lumens, solid nests or sheets of tumor cells, trabecular arrangements, or single-file infiltration.

Task Relevance:

- Determining whether a region is normal or invasive is the most fundamental and clinically meaningful decision in breast pathology.
- For machine learning models, highly discriminative cues include cell density and pleomorphism, architectural irregularity, loss of the myoepithelial layer, desmoplastic stroma, and abnormal gland formation.

DCIS/IC prompt.

You are assisting with a breast pathology image classification task. The goal is to distinguish ductal carcinoma in situ (DCIS) from invasive carcinoma (IC) in H&E-stained microscopy images. Below is the clinical and morphological context you must rely on.

Ductal Carcinoma In Situ (DCIS):

- DCIS consists of malignant epithelial cells that remain confined within the mammary ducts.
- The basement membrane is intact, and a myoepithelial cell layer surrounds the ducts. This intact myoepithelial layer is a key diagnostic hallmark.
- Tumor cells fill or partially fill ducts without extending into the surrounding stroma.
- Ducts may be dilated or distended.
- Architectural patterns include comedo, cribriform, solid, papillary, and micropapillary.
- Cytologic atypia is present but restricted to intraductal spaces.
- Comedo necrosis may be seen.
- No desmoplastic stromal reaction is present because invasion has not occurred.

Invasive Carcinoma (IC):

- IC is defined by malignant epithelial cells breaching the basement membrane and infiltrating the surrounding stroma.
- The myoepithelial layer is absent around invasive tumor structures.
- Tumor growth appears as irregular glands, nests, cords, or single tumor cells invading the fibrous stroma.
- Normal duct architecture is disrupted or destroyed.
- Often accompanied by desmoplastic stromal reaction (fibrotic response).
- Increased cytologic atypia and mitotic activity.
- Common invasive patterns include abnormal tubule formation, solid nests, trabecular arrangements, or single-file infiltration.

Key Distinguishing Criteria:

- DCIS: intact basement membrane, preserved myoepithelial layer, malignant cells confined within ducts, no stromal invasion, no desmoplastic reaction.
- IC: basement membrane breached, myoepithelial layer absent, tumor cells infiltrating stroma, architectural destruction, desmoplastic response present.

N/DCIS/IC prompt.

You are assisting with a breast pathology image classification task. The goal is to distinguish among three categories in H&E-stained microscopy images: Normal breast tissue (N), Ductal Carcinoma in Situ (DCIS), and Invasive Carcinoma (IC). Use the following clinical and morphological definitions to ground your reasoning.

Normal Breast Tissue (N):

- Normal tissue consists of terminal duct–lobular units (TDLUs) composed of small acini arranged into lobules.
- Ducts and acini have a dual-layer structure: an inner luminal epithelial cell layer and an outer myoepithelial cell layer.
- The basement membrane is intact and undisturbed.
- Architecture is well-organized, with uniform nuclei, low atypia, low mitotic figures, and no stromal desmoplasia.
- Surrounding stroma contains a mixture of fibrous connective tissue and adipose tissue.

Ductal Carcinoma in Situ (DCIS):

- DCIS is a non-invasive proliferation of malignant epithelial cells confined within the mammary ducts.
- The basement membrane remains intact, and the myoepithelial layer is preserved around ductal structures.
- Tumor cells fill or partially fill the ducts without breaching the basement membrane.
- Ducts may be dilated or expanded.
- Architectural patterns may include comedo, cribriform, solid, papillary, or micropapillary growth.
- Cytologic atypia is present, and comedo necrosis may be observed.
- No invasion into the surrounding stroma and no desmoplastic reaction.

Invasive Carcinoma (IC):

- IC is defined by malignant epithelial cells that have breached the basement membrane and infiltrated into the surrounding stroma.
- The myoepithelial cell layer is absent around invasive tumor structures.
- Growth forms include irregular tubules, nests, cords, sheets, or single tumor cells infiltrating fibrous stroma.
- Native duct/lobular architecture is disrupted or destroyed.
- Often associated with desmoplastic stromal reaction (fibrotic response), increased cytologic atypia, and mitotic activity.

Key Distinguishing Criteria:

- Normal (N): intact architecture with dual epithelial+myoepithelial layers, no atypical proliferation, no stromal invasion.
- DCIS: malignant cells confined within ducts; intact basement membrane and myoepithelial layer; no stromal invasion or desmoplasia.
- IC: basement membrane breached; no myoepithelial layer; tumor cells infiltrating stroma with desmoplastic reaction and loss of normal architecture.

Using this information, when given an image, identify which morphological features are present and determine whether the image belongs to Normal tissue, DCIS, or Invasive Carcinoma.

Major issue 3:

It is not clear to me why the notion of generalization is used, given that the pipeline is not fitted to a specific dataset. It would however be interesting to see how well the generated prompts generalize across samples and across models. If these generalize well, the optimized prompts could be made available for use in this context.

Response:

We thank the reviewer for this insightful observation regarding our terminology and for the excellent suggestion to evaluate cross-model prompt transferability. We agree that the term "generalization" could be conflated with statistical fitting in traditional supervised learning. Accordingly, we have revised the manuscript to use more precise terms such as "adaptability," "versatility," and "applicability" when referring to the framework's capacity to operate across different datasets and models (Abstract; Page 7; Figure 5).

Furthermore, following the reviewer's suggestion, we evaluated the transferability of optimized prompts across models. We conducted experiments where prompts optimized by one LLM (e.g., GPT-4o) were used for inference by a different LLM (e.g., Gemini 2.0 Flash). As shown in the revised Table 1, this cross-model application yielded substantial accuracy gains over unoptimized baselines. Notably, applying GPT-4o-optimized criteria to Gemini 2.0 Flash achieved 86.75% accuracy, demonstrating that the diagnostic logic discovered by our framework captures transferable morphological principles rather than model-specific artifacts. These results and the corresponding discussion have been added to Page 7 and the Supplementary Information.

Revisions:

1. Page 9, start of the subsection "RECAP-PATH generalizes across datasets..."
Original: "Complementing our cross-model analysis, we next evaluate **cross-dataset generalization...**"
Revised: "Complementing our cross-model analysis, we next evaluate **the adaptability of our framework to additional histopathology datasets.**"
2. Page 7, in the section "RECAP-PATH enables interpretable subtype classification in breast cancer pathology", a paragraph has been added:
"We also assessed prompt transferability. we applied prompts optimized by GPT-4o to Gemini 2.0 Flash for inference, and vice versa (Table 1). This cross-model application yielded substantial accuracy gains over unoptimized baselines; for instance, applying GPT-4o-optimized criteria to Gemini 2.0 Flash achieved 86.8% accuracy (compared to 71% baseline)."

While the highest performance was consistently achieved when the optimization and inference models were identical, these results confirm that the generated diagnostic logic captures transferable morphological principles rather than model-specific artifacts.”

3. Abstract:

We have revised the abstract format to comply with the journal’s specific requirements and removed the use of the word “generalize.”

4. Page 7:

Original: "Lastly, we assessed robustness and model-agnostic **generalization**."

Revised: "We then assessed robustness and model-agnostic **versatility**."

5. Table 1 caption:

Original: "...demonstrating the model-agnostic **generalizability** of RECAP-PATH."

Revised: "...demonstrating the model-agnostic **applicability** of RECAP-PATH."

6. Figure 5 caption:

Original: "Figure 5. **Generalization** of RECAP-PATH across pathology datasets."

Revised: "Figure 5. **Adaptability** of RECAP-PATH across pathology datasets."

7. The result below has been added to Table 1.

Training	Categories	Inference Model	Acc	Std	Upper	Lower
GPT-4o train	N/IC	gemini 2.0 001 flash	0.8675	0.0218	0.8978	0.8372
Gemini 2.0 train	N/IC	gpt-4o-2024-05-13	0.7096	0.0190	0.7359	0.6832

Minor issue 1:

Figures 2D, 4E, 5A, 5C: Add axis labels to the confusion matrices.

Revision:

We have added axis labels to all the confusion matrices, including Figs. 2D, 4A, 4E, 5A, and 5C.

Minor issue 2:

Table 1: Consider merging accuracy and its confidence interval into a single line: Acc (lower CI, upper CI).

Revision:

We thank the Reviewer for this suggestion, which has now been implemented.

Minor issue 3:

Typos / Wording: Page 1: “Bu -> But”

Revision:

The typo has now been corrected.

Reviewer's conclusion:

I enjoyed reading the paper and it presents an interesting approach. I would recommend to address the issues that I have raised above to improve the value of the manuscript for the scientific community and to substantiate the key claim regarding interpretability and clinical utility.

Response:

We thank the reviewer for their encouraging conclusion and for identifying specific areas where our claims required greater precision and validation. We believe that by reframing our contribution around "audit-ready" diagnostics rather than intrinsic interpretability, and by rigorously benchmarking performance against supervised and in-context baselines, we have significantly enhanced the scientific rigor and clarity of the manuscript. We are confident that these revisions fully address the raised concerns and substantiate the clinical utility of the RECAP-PATH framework for the broader scientific community.

Reviewer #2

The paper presents RECAP-PATH, a framework for interpretable diagnostic reasoning in digital pathology using a two-phase prompt optimization strategy with multimodal large language models (MLLMs). The method enables pathology-style description generation using general-purpose models without finetuning, showing strong accuracy on datasets such as BACH and SICAP. The study is well evaluated, with convincing results and validation by board-certified pathologists.

A. Strengths:

1. Novel prompt optimization strategy: The two-phase optimization (diversification + refinement) is well-motivated and effectively balances reasoning diversity and diagnostic accuracy. It is conceptually elegant and clearly presented.
2. Experiment design: The experiments are well structured and anticipate the reader's questions, showing strong experimental intuition.
3. Expert evaluation: The inclusion of three board-certified pathologists and a human-in-the-loop setup adds valuable external validation and clinical credibility.

Response:

We sincerely thank the Reviewer for their thorough evaluation and encouraging feedback. We are particularly grateful for the recognition of our two-phase optimization strategy as "conceptually elegant" and "well-motivated," as well as the appreciation for our experimental design and the inclusion of board-certified pathologists for clinical validation.

We have carefully addressed all the constructive suggestions raised by the Reviewer. Specifically, we have strengthened the manuscript by:

- **Benchmarking against Prompt Optimization Baselines (Major Concern 1):** We compared RECAP-PATH against Automatic Prompt Engineering (APE), demonstrating that our reasoning-driven feedback loop significantly outperforms standard selection-based optimization. We also added supervised and manual prompting baselines to better contextualize performance.
- **Validating on Open-Source Models (Major Concern 2):** To address data privacy concerns, we successfully validated our framework on the locally deployable Qwen2.5-VL-72B-Instruct model, confirming that RECAP-PATH is adaptable to open-weight architectures.
- **Contextualizing within Related Work (Major Concern 3):** We significantly expanded the Discussion section to survey the landscape of prompt optimization, contrasting search-based methods (like APE) with feedback-driven strategies.
- **Refining Visuals and Text (Minor B.2 & C):** We improved the clarity of the Figure 2 captions and corrected all identified grammatical errors.

We believe these revisions have significantly improved the robustness and completeness of our study. A point-by-point response to each specific comment follows below.

B. Weaknesses:

Major concern 1:

Prompt optimization baselines: Compare RECAP-PATH against other prompt optimization methods to demonstrate its advantages.

Response:

We thank the reviewer for this insightful suggestion. To benchmark RECAP-PATH against established prompt optimization methods, we implemented Automatic Prompt Engineering (APE; Zhou et al., 2022) as a baseline.

As shown in the table below, RECAP-PATH demonstrated improved performance compared to APE on the Normal vs. Invasive Carcinoma (N/IC) task using Gemini 2.0 Flash (91.25% vs. 88.00%). We attribute this difference to the distinct optimization mechanisms employed by each framework. APE primarily relies on generating and selecting candidate prompts but lacks an iterative feedback loop where an LLM strategically revises the prompt based on performance. In contrast, RECAP-PATH utilizes a reasoning model to explicitly reflect on errors and employs a multi-arm optimization strategy. This allows our framework to systematically refine the diagnostic criteria through iterative reasoning, leading to superior performance compared to the selection-based approach of APE.

Training	Categories	Acc	Std	Upper	Lower
gemini 2.0 001 flash	N/IC	0.8800	0.0350	0.9286	0.8314

Beyond APE, we have further contextualized RECAP-PATH’s performance against both supervised and manual prompting baselines. We established a 'performance ceiling' using a supervised linear classifier trained on CLIP embeddings (~92.5% accuracy) and a 'static expert' baseline using manual in-context learning (86.0% accuracy). RECAP-PATH (91.25%) effectively bridges this gap, significantly outperforming static expert prompts and achieving near-supervised accuracy without the need for parameter updates or opaque feature embeddings. These results have been added to Table 1, with corresponding discussion included in the appropriate sections.

Revisions:

1. We added a paragraph on Page 5, within the “Two-Phase Prompt Optimization...” subsection, that comprehensively describes the benchmarking results.

*“To benchmark this performance, we established a supervised upper bar using a linear classifier trained on Contrastive Language-Image Pre-training (CLIP) embeddings ~\cite{radford2021learning}, which achieved 92.5\% accuracy. ... RECAP-PATH attains comparable results without requiring weight updates or white-box access, while providing pathologist-readable rationales. Additionally, an in-context learning baseline using manually curated clinical definitions yielded lower performance (86.0\% for N/IC) than RECAP-PATH (Table 1, Fig. S2). **We further benchmarked against Automatic Prompt Engineering (APE) ~\cite{zhou2022large}, which achieved 88.0\% (95\% CI: 83.1\%–92.9\%) and underperformed RECAP-PATH (91.3\%). These comparisons indicate that neither static expert knowledge nor general-purpose instruction tuning is sufficient without the systematic, reasoning-driven alignment provided by our optimization.**”*

2. We have added a paragraph in the “Comparison methods and validation benchmarks” subsection of the Methods describing our use of APE as a baseline.

“\item \textbf{APE} \cite{zhou2022large} (Automatic Prompt Engineering) represents a state-of-the-art in automated prompt optimization. APE frames prompt engineering as a black-box optimization problem where instructions are treated as programs to be generated and selected by an LLM. The framework starts with a pool of candidate prompts generated by an

LLM, evaluates them based on zero-shot task accuracy or similar metrics, and iteratively refines the selection. We used this baseline to compare our proposed method against a pure LLM-driven prompt search approach that does not explicitly leverage the structured visual description loop used in our pipeline.”

Major concern 2:

Open-source models: Since data privacy is critical in the medical domain, it would be valuable to test RECAP-PATH with open-source models that can be run locally (e.g. MedGemma, Qwen-VL).

Response:

We appreciate the suggestion to validate our framework on open-source architectures. To address this, we applied RECAP-PATH to Qwen2.5-VL-72B-Instruct, a leading open-source model suitable for local deployment. Due to the excessive computational turnaround time required when utilizing local resources for this model, we focused this analysis on the binary classification case (Normal vs. Invasive Carcinoma) as a proof-of-concept validation.

Within this scope, the results confirm our framework’s model-agnostic adaptability: optimization improved diagnostic accuracy from an initial 55.50% (95% CI: 49.84%–61.16%) to 67.00% (95% CI: 62.53%–71.47%). This significant improvement demonstrates that RECAP-PATH successfully drives reasoning gains in open-weight models. While the absolute performance currently trails larger proprietary models, these findings validate that our method is compatible with privacy-preserving, locally deployable workflows.

Training model	Task	Prompt	Acc	Std	Upper	Lower
Qwen2.5-VL-72B-Instruct	N/IC	Initial	0.5550	0.0408	0.6116	0.4984
Qwen2.5-VL-72B-Instruct	N/IC	Optimized	0.6700	0.0322	0.7147	0.6253

Revisions:

1. We added a paragraph describing this result in the subsection “RECAP-PATH enables interpretable subtype classification in breast cancer pathology”:

“Lastly, to address clinical data privacy requirements, we validated the framework on the open-source Qwen2.5-VL-72B-Instruct model. Focusing on binary classification (Normal vs. Invasive Carcinoma) as a proof-of-concept due to excessive local computational turnaround times, RECAP-PATH improved diagnostic accuracy from 55.5% (95% CI: 49.8%–61.2%) to 67.0% (95% CI: 62.5%–71.5%), demonstrating its compatibility with locally deployable, privacy-preserving workflows.”

2. We have added a paragraph in the “Comparison methods and validation benchmarks” subsection of the Methods describing our use of Qwen2.5-VL to evaluate the applicability of our framework to open-source models.

“\item \textbf{Qwen2.5-VL} \cite{bai2023qwen} serves as a representative baseline for MLLMs. It integrates a native dynamic-resolution Vision Transformer~\cite{dosovitskiy2020image} with the Qwen2.5 language model. The architecture incorporates enhancements such as window attention, SwiGLU activations, RMSNorm~\cite{zhang2019root}, and Multimodal Rotary Position Embeddings~\cite{heo2024rotary} to efficiently model spatial and temporal information. We

utilized the instruction-tuned variant, which is optimized for visual reasoning and instruction following, to evaluate performance when a single, unified model processes both visual and textual inputs without the explicit two-phase optimization we propose.”

Major concern 3:

Related work: Expand the discussion of prior prompt optimization strategies to provide clearer context for the proposed approach.

Response:

We thank the reviewer for highlighting the need to better contextualize RECAP-PATH within the rapidly evolving field of prompt optimization. In response, we have significantly expanded the Discussion section (Page 10) to provide a comprehensive literature survey. We now explicitly categorize state-of-the-art strategies into search-based methods (e.g., APE, OPRO, PromptAgent) and feedback-driven methods (e.g., APO, TextGrad, Reflexion).

We have expanded our discussion to better contextualize the specific trade-offs of these approaches within the pathology domain. Our supplementary benchmarking suggests that general search-based methods, such as APE, yielded slightly lower classification accuracy in this context compared to our framework. Furthermore, while feedback-driven strategies like APO are highly effective, their primary focus on maximizing scalar accuracy metrics risks encouraging 'shortcut learning', where the model may converge on concise prompts that optimize performance scores, potentially at the expense of the step-by-step reasoning required for clinical auditability. We have revised the text to clarify that RECAP-PATH's 'diversify-to-optimize' strategy aims to address this balance, seeking to achieve competitive accuracy while preserving the descriptive granularity essential for trust.

Revisions:

We added a paragraph to the Discussions section. The inserted paragraph reads:

“Parallel to these adaptation strategies, recent work has formalized prompt engineering into an automated optimization problem. Search-based methods, most notably APE ~\cite{zhou2022large}, Optimization by PROMpting (OPRO) ~\cite{yang2023large}, and PromptAgent ~\cite{wang2023promptagent}, leverage LLMs to iteratively generate and select instructions. However, our supplementary benchmarking suggests that broad search strategies like APE yielded slightly lower classification accuracy in this context compared to our framework. Distinct from search, feedback-driven frameworks like Automatic Prompt Optimization (APO) ~\cite{pryzant2023automatic}, TextGrad ~\cite{yuksekgonul2024textgrad}, and Reflexion ~\cite{shinn2023reflexion} mimic gradient descent by using textual critiques to refine prompts. While effective, their primary focus on maximizing scalar accuracy metrics risks encouraging shortcut learning, converging on concise cues that optimize scores potentially at the expense of the step-by-step reasoning necessary for clinical auditability. RECAP-PATH diverges from these approaches with a self-learning, model-agnostic framework that prioritizes a 'diversify-to-optimize' axis. Rather than optimizing solely for the final label, it aims to address this balance by refining diagnosis criteria to yield evidence-linked image descriptions, ensuring the model articulates morphological features before prediction.”

B.2. Minor concern:

1. Figure 2: The gray samples are unclear at first sight. Improve the visualization or caption for better readability.

Revisions:

We have revised the Fig. 2 caption to define the gray points: "... (F) UMAP visualization of the description embeddings. **Gray points represent aggregate embeddings from all stages as a static reference. Initially, descriptions ...**"

B.3. Optional

1. Baselines: Include a comparison to multimodal pathology models such as Quilt-LLaVA to better contextualize performance.

Response:

We thank the Reviewer for this insightful suggestion. We agree that comparing performance against specialized multimodal pathology models is an interesting direction. To explore this, we conducted a preliminary evaluation of Quilt-LLaVA (v1.5-7b checkpoint) on our dataset (Normal vs. Invasive Carcinoma) in a zero-shot setting. However, we observed that without extensive prompt engineering or fine-tuning specific to our task, the model exhibited severe class collapse (predicting a single class for all samples).

Investigating the root cause of this failure and optimizing the prompts to achieve a fair and representative baseline performance would require a significant diversion from the current study's scope. Furthermore, simply reporting the failed zero-shot performance might be misleading regarding the potential of such models. Given that this request was optional, and to maintain the focus of our manuscript on the proposed framework, we have decided not to include this specific comparison in the revised text.

C. Grammar/Typos

1. Typo: "Image description generation and diganosis" (p. 11).

2. Writing error: "To assess the clinical relevance and interpretability of RECAP-PATH, we conducted a blind evaluation survey involving three board-certified pathologists. with images and their corresponding descriptions generated by MLLM."

Revisions:

Both errors have now been corrected.

Revised: "Image description generation and diagnosis"

Revised: "To assess the clinical relevance and interpretability of RECAP-PATH, we conducted a blind evaluation survey in which three board-certified pathologists reviewed images and their corresponding descriptions generated by the MLLM."